# Lookahead Counterfactual Fairness

**Zhiqun Zuo**                                                   *zuo.167@osu.edu*
*Department of Computer Science and Engineering*
*The Ohio State University*

**Tian Xie**                                                     *xie.1379@osu.edu*
*Department of Computer Science and Engineering*
*The Ohio State University*

**Xuwei Tan**                                                    *tan.1206@osu.edu*
*Department of Computer Science and Engineering*
*The Ohio State University*

**Xueru Zhang**                                                  *zhang.12807@osu.edu*
*Department of Computer Science and Engineering*
*The Ohio State University*

**Mohammad Mahdi Khalili**                                       *khalili.17@osu.edu*
*Department of Computer Science and Engineering*
*The Ohio State University*

**Reviewed on OpenReview:** *https://openreview.net/forum?id=kmHq3pKIlj&referrer=%5Bthe%20profile%20of%20Zhiqun%20Zuo%5D(%2Fprofile%3Fid%3D~Zhiqun_Zuo1)*

## Abstract

As machine learning (ML) algorithms are used in applications that involve humans, concerns have arisen that these algorithms may be biased against certain social groups. *Counterfactual fairness* (CF) is a fairness notion proposed in Kusner et al. (2017) that measures the unfairness of ML predictions; it requires that the prediction perceived by an individual in the real world has the same marginal distribution as it would be in a counterfactual world, in which the individual belongs to a different group. Although CF ensures fair ML predictions, it fails to consider the downstream effects of ML predictions on individuals. Since humans are strategic and often adapt their behaviors in response to the ML system, predictions that satisfy CF may not lead to a fair future outcome for the individuals. In this paper, we introduce *lookahead counterfactual fairness* (LCF), a fairness notion accounting for the downstream effects of ML models which requires the individual *future status* to be counterfactually fair. We theoretically identify conditions under which LCF can be satisfied and propose an algorithm based on the theorems. We also extend the concept to path-dependent fairness. Experiments on both synthetic and real data validate the proposed method[1].

## 1 Introduction

The integration of machine learning (ML) into high-stakes domains (e.g., lending, hiring, college admissions, healthcare) has the potential to enhance traditional human-driven processes. However, it may introduce the risk of perpetuating biases and unfair treatment of protected groups. For instance, the violence risk assessment tool SAVRY has been shown to discriminate against males and foreigners (Tolan et al., 2019); Amazon's previous hiring system exhibited gender bias (Dastin, 2018); the accuracy of a computer-aided clinical diagnostic system varies significantly across patients from different racial groups (Daneshjou et al.,

---

[1]The code for this paper is available in `https://github.com/osu-srml/LCF`.

2021). Numerous fairness notions have been proposed in the literature to address unfairness issues, including *unawareness fairness* that prevents the explicit use of demographic attributes in the decision-making process, *parity-based fairness* that equalizes certain statistics (e.g., accuracy, true/false positive rate) across different groups (Hardt et al., 2016b; Khalili et al., 2023; 2021b;a; Abroshan et al., 2024), *preference-based fairness* that ensures a group of individuals, as a whole, regard the results or consequences they receive from the ML system more favorably than those received by another group (Zafar et al., 2017; Do et al., 2022). Unlike these notions that overlook the underlying causal structures among different variables Kusner et al. (2017) introduced the concept of *counterfactual fairness* (CF), which requires that an individual should receive a consistent treatment distribution in a counterfactual world where their sensitive attributes differs. Since then many approaches have been developed to train ML models that satisfy CF (Chiappa, 2019; Zuo et al., 2022; Wu et al., 2019; Xu et al., 2019; Ma et al., 2023; Zuo et al., 2023; Abroshan et al., 2022).

However, CF is primarily studied in static settings without considering the downstream impacts ML decisions may have on individuals. Because humans in practice often adapt their behaviors in response to the ML system, their future status may be significantly impacted by ML decisions (Miller et al., 2020; Shavit et al., 2020; Hardt et al., 2016a). For example, individuals receiving approvals in loan applications may have more resources and be better equipped to improve their future creditworthiness (Zhang et al., 2020). Content recommended in digital platforms can steer consumer behavior and reshape their preferences (Dean & Morgenstern, 2022; Carroll et al., 2022). As a result, a model that satisfies CF in a static setting without accounting for such downstream effects may lead to unexpected adverse outcomes.

Although the downstream impacts of fair ML have also been studied in prior works (Henzinger et al., 2023a; Xie & Zhang, 2024a; Ge et al., 2021; Henzinger et al., 2023b; Liu et al., 2018; Zhang et al., 2020; Xie et al., 2024), the impact of counterfactually fair decisions remain relatively unexplored. The most related work to this study is (Hu & Zhang, 2022), which considers sequential interactions between individuals and an ML system over time and their goal is to ensure ML *decisions* satisfy path-specific counterfactual fairness constraint throughout the sequential interactions. However, Hu & Zhang (2022) still focuses on the fairness of ML decisions but not the fairness of the individual's actual status. Indeed, it has been well-evidenced that ML decisions satisfying certain fairness constraints during model deployment may reshape the population and unintentionally exacerbate the group disparity (Liu et al., 2018; Zhang et al., 2019; 2020). A prime example is Liu et al. (2018), which studied the lending problem and showed that the lending decisions satisfying statistical parity or equal opportunity fairness (Hardt et al., 2016b) may actually cause harm to disadvantaged groups by lowering their future credit scores, resulting in amplified group disparity. Tang et al. (2023) considered sequential interactions between ML decisions and individuals, where they studied the impact of counterfactual fair predictions on statistical fairness but their goal is still to ensure parity-based fairness at the group level.

In this work, we focus on counterfactual fairness evaluated over individual *future* status (label), which accounts for the downstream effects of ML decisions on individuals. We aim to examine under what conditions and by what algorithms the disparity between individual future status in factual and counterfactual worlds can be mitigated after deploying ML decisions. To this end, we first introduce a new fairness notion called "lookahead counterfactual fairness (LCF)." Unlike the original counterfactual fairness proposed by Kusner et al. (2017) that requires the ML predictions received by individuals to be the same as those in the counterfactual world, LCF takes one step further by enforcing the individual future status (after responding to ML predictions) to be the same.

Given the definition of LCF, we then develop algorithms that learn ML models under LCF. To model the effects of ML decisions on individuals, we focus on scenarios where individuals subject to certain ML decisions adapt their behaviors strategically by increasing their chances of receiving favorable decisions; this can be mathematically formulated as modifying their features toward the direction of the gradient of the decision function (Rosenfeld et al., 2020; Xie & Zhang, 2024b). We first theoretically identify conditions under which an ML model can satisfy LCF, and then develop an algorithm for training ML models under LCF. We also extend the algorithm and theorems to path-dependent LCF, which only considers unfairness incurred by the causal effect from the sensitive attribute to the outcome along certain paths.

Our contributions can be summarized as follows:

- We propose lookahead counterfactual fairness (LCF), a novel fairness notion that evaluates counterfactual fairness over individual future status (i.e., actual labels after responding to ML systems). Unlike the original CF notion that focuses on current ML predictions, LCF accounts for the subsequent impacts of ML decisions and aims to ensure fairness over individual actual future status. We also extend the definition to path-dependent LCF.

- For scenarios where individuals respond to ML models by changing features toward the direction of the gradient of decision functions, we theoretically identify conditions under which an ML model can satisfy LCF. We further develop an algorithm for training ML models under LCF.

- We conduct extensive experiments on both synthetic and real data to validate the proposed algorithm. Results show that compared to conventional counterfactual fair predictors, our method can improve disparity with respect to the individual actual future status.

## 2 Related Work

Causal fairness has been explored in many aspects in recent years' research. Kilbertus et al. (2017) point out that no observational criterion can distinguish scenarios determined by different causal mechanisms but have the same observational distribution. They propose the definition of unresolved discrimination and proxy discrimination based on the intuition that some of the paths from the sensitive attribute to the prediction can be acceptable. Nabi & Shpitser (2018) argue that a fair causal inference on the outcome can be obtained by solving a constrained optimization problem. These notions are based on defining constraints on the interventional distributions.

Counterfactual fairness (Kusner et al., 2017) requires the prediction on the target variable to have the same distribution in the factual world and counterfactual world. Many extensions of traditional statistical fairness notions such as Fair on Average Causal Effect (FACE) (can be regarded as counterfactual demographic parity) (Khademi et al., 2019), Fair on Average Causal Effect on the Treated (FACT) (can be regarded as counterfactual equalized odds) (Khademi et al., 2019), and CAPI fairness (counterfactual individual fairness) (Ehyaei et al., 2024) have been proposed. Path-specific counterfactual fairness (Chiappa, 2019) considered the path-specific causal effect. However, the notions are focused on fairness in static settings and do not consider the future effect. Most recent work about counterfactual fairness is about achieving counterfactual fairness in different applications, such as graph data (Wang et al., 2024a;b), medical LLMs (Poulain et al., 2024), or software debuging (Xiao et al., 2024). Some literature try to use counterfactual fairness for explanations (Goethals et al., 2024). Connecting counterfactual fairness with group fairness notions (Anthis & Veitch, 2024) or exploring counterfactual fairness with partial knowledge about the causal model (Shao et al., 2024; Pinto et al., 2024; Duong et al., 2024; Zhou et al., 2024) are also receiving much attention. Machado et al. (2024) propose an idea of interpretable counterfactual fairness by deriving counterfactuals with optimal transports (De Lara et al., 2024). While extending the definition of counterfactual fairness to include the downstream effects has been less focused on.

Several studies in the literature consider the downstream effect on fairness of ML predictions. There are two kinds of objectives in the study of downstream effects: ensuring fair predictions in the future or fair true status in the future. The two most related works to our paper are Hu & Zhang (2022) and Tang et al. (2023). Hu & Zhang (2022) consider the problem of ensuring ML predictions satisfy path-specific counterfactual fairness over time after interactions between individuals and an ML system. Tang et al. (2023) study the impact on the future true status of the ML predictions. Even though they considered the impact of a counterfactually fair predictor, their goal is to ensure parity-based fairness. Therefore, the current works lack the consideration of ensuring counterfactual fairness on the true label after the individual responds to the current ML prediction. Our paper is aimed at solving this problem.

## 3 Problem Formulation

Consider a supervised learning problem with a training dataset consisting of triples $(A, X, Y)$, where $A \in \mathcal{A}$ is a sensitive attribute distinguishing individuals from multiple groups (e.g., race, gender),

$X = [X_1, X_2, ..., X_d]^{\mathrm{T}} \in \mathcal{X}$ is a $d$-dimensional feature vector, and $Y \in \mathcal{Y} \subseteq \mathbb{R}$ is the target variable indicating individual's underlying status (e.g., $Y$ in lending identifies an applicant's ability to repay the loan, $Y$ in healthcare may represent patients' insulin spike level). The goal is to learn a predictor from training data that can predict $Y$ given inputs $A$ and $X$. Let $\hat{Y}$ denote the output of the predictor.

We assume $(A, X, Y)$ is associated with a structural causal model (SCM) (Pearl et al., 2000) $\mathcal{M} = (V, U, F)$, where $V = (A, X, Y)$ represents observable variables, $U$ includes unobservable (exogenous) variables that are not caused by any variable in $V$, and $F = \{f_1, f_2, \ldots, f_{d+2}\}$ is a set of $d+2$ functions called *structural equations* that determines how each observable variable is constructed. More precisely, we have the following structural equations,

$$
\begin{aligned}
X_i &= f_i(pa_i, U_{pa_i}), \ \forall i \in \{1, \cdots, d\}, \\
A &= f_A(pa_A, U_{pa_A}), \\
Y &= f_Y(pa_Y, U_{pa_Y}),
\end{aligned}
\tag{1}
$$

where $pa_i \subseteq V$, $pa_A \subseteq V$ and $pa_Y \subseteq V$ are observable variables that are the parents of $X_i$, $A$, and $Y$, respectively. $U_{pa_i} \subseteq U$ are unobservable variables that are the parents of $X_i$. Similarly, we denote unobservable variables $U_{pa_A} \subseteq U$ and $U_{pa_Y} \subseteq U$ as the parents of $A$ and $Y$, respectively.

### 3.1 Background: counterfactuals

If the probability density functions of unobserved variables are known, we can leverage the structural equations in SCM to find the marginal distribution of any observable variable $V_i \in V$ and even study how intervening certain observable variables impacts other variables. Specifically, the **intervention** on variable $V_i$ is equivalent to replacing structural equation $V_i = f_i(pa_i, U_{pa_i})$ with equation $V_i = v$ for some $v$. Given new structural equation $V_i = v$ and other unchanged structural equations, we can find out how the distribution of other observable variables changes as we change value $v$.

In addition to understanding the impact of an intervention, SCM can further facilitate **counterfactual inference**, which aims to answer the question "*what would be the value of $Y$ if $Z$ had taken value $z$ in the presence of evidence $O = o$ (both $Y$ and $Z$ are two observable variables)?*" The answer to this question is denoted by $Y_{Z \leftarrow z}(U)$ with $U$ following conditional distribution of $\Pr\{U = u | O = o\}$. Given $U = u$ and structural equations $F$, the counterfactual value of $Y$ can be computed by replacing the structural equation of $Z$ with $Z = z$ and replacing $U$ with $u$ in the rest of the structural equations. Such counterfactual is typically denoted by $Y_{Z \leftarrow z}(u)$. Given evidence $O = o$, the distribution of counterfactual value $Y_{Z \leftarrow z}(U)$ can be calculated as follows,[2]

$$
\Pr\{Y_{Z \leftarrow z}(U) = y | O = o\} = \sum_u \Pr\{Y_{Z \leftarrow z}(u) = y\} \Pr\{U = u | O = o\}.
\tag{2}
$$

**Example 3.1 (Law school success).** Consider two groups of college students distinguished by gender $A \in \{0, 1\}$ whose first-year average (FYA) in college is denoted by $Y$. The FYA of each student is causally related to (observable) grade-point average (GPA) before entering college $X_G$, entrance exam score (LSAT) $X_L$, and gender $A$. Suppose there are two unobservable variables $U = (U_A, U_{XY})$, e.g., $U_{XY}$ may be interpreted as the student's knowledge. Consider the following structural equations:

$$
\begin{aligned}
A &= U_A, & X_G &= b_G + w_G^A A + U_{XY}, \\
X_L &= b_L + w_L^A A + U_{XY}, & Y &= b_F + w_F^A A + U_{XY},
\end{aligned}
$$

where $(b_G, w_G^A, b_L, w_L^A, b_F, w_F^A)$ are know parameters of the causal model. Given observation $X_G = 1, A = 0$, the counterfactual value can be calculated with an *abduction-action-prediction* procedure Glymour et al. (2016): (i) *abduction* that finds posterior distribution $\Pr\{U = u | X_G = 1, A = 0\}$. Here, we have $U_{XY} = 1 - b_G$ and $U_A = 0$ with probability 1; (ii) *action* that performs intervention $A = 1$ by replacing structural equations

---

[2]Given structural equations equation 1 and the marginal distribution of $U$, $\Pr\{U = u, O = o\}$ can be calculated using the Change-of-Variables Technique and the Jacobian factor. As a result, $\Pr\{U = u | O = o\} = \frac{\Pr\{U = u, O = o\}}{\Pr\{O = o\}}$ can also be calculated.

of $A$; (iii) *prediction* that computes distribution of $Y_{A \leftarrow 1}(U)$ given $X_G = 1, A = 0$ using new structural equations and the posterior. We have:

$$Y_{A \leftarrow 1}(U) = b_f + w_F^A + 1 - b_G \quad \text{with probability 1.}$$

## 3.2 Counterfactual Fairness

Counterfactual Fairness (CF) was first proposed by Kusner et al. (2017); it requires that for an individual with $(X = x, A = a)$, the prediction $\hat{Y}$ in the factual world should be the same as that in the counterfactual world in which the individual belongs to a different group. Mathematically, CF is defined as follows: $\forall a, \breve{a} \in \mathcal{A}, X \in \mathcal{X}, y \in \mathcal{Y}$,

$$\Pr\left(\hat{Y}_{A \leftarrow a}(U) = y | X = x, A = a\right) = \Pr\left(\hat{Y}_{A \leftarrow \breve{a}}(U) = y | X = x, A = a\right),$$

While the CF notion has been widely used in the literature, it does not account for the downstream impacts of ML prediction $\hat{Y}$ on individuals in factual and counterfactual worlds. To illustrate the importance of considering such impacts, we provide an example below.

**Example 3.2.** Consider automatic lending where an ML model is used to decide whether to issue a loan to an applicant based on credit score $X$ and sensitive attribute $A$. As highlighted in Liu et al. (2018), issuing loans to unqualified people who cannot repay the loan may hurt them by worsening their future credit scores. Assume an applicant in the factual world is qualified for the loan and does not default. But in a counterfactual world where the applicant belongs to another group, he/she is not qualified. Under counterfactually fair predictions, both individuals in the factual and counterfactual worlds should receive the loan with the same probability. Suppose both are issued a loan, then the one in the counterfactual world would have a worse credit score in the future. Thus, it is crucial to consider the downstream effects when learning a fair ML model.

## 3.3 Characterize downstream effects

Motivated by Example 3.2, this work studies CF in a dynamic setting where the deployed ML decisions may affect individual behavior and change their future features and statuses. Formally, let $X'$ and $Y'$ denote an individual's future feature vector and status, respectively. We use an individual response $r$ to capture the impact of ML prediction $\hat{Y}$ on individuals, as defined below.

**Definition 3.1** (Individual response). An individual response $r : \mathcal{U} \times \mathcal{V} \times \mathcal{Y} \mapsto \mathcal{U} \times \mathcal{V}$ is a map from the current exogenous variables $U \in \mathcal{U}$, endogenous variables $V \in \mathcal{V}$, and prediction $\hat{Y} \in \mathcal{Y}$ to the future exogenous variables $U'$ and endogenous variables $V'$.

One way to tackle the issue in Example 3.2 is to explicitly consider the individual response and impose a fairness constraint on future status $Y'$ instead of the prediction $\hat{Y}$. We call such a fairness notion the *Lookahead Counterfactual Fairness (LCF)* and present it in Section 4.

## 4 Lookahead Counterfactual Fairness

We consider the fairness over the individual's future outcome $Y'$. Given structural causal model $\mathcal{M} = (U, V, F)$, individual response $r$, and data $(A, X, Y)$, we define lookahead counterfactual fairness below.

**Definition 4.1.** We say an ML model satisfies lookahead counterfactual fairness (LCF) under a response $r$ if the following holds $\forall a, \breve{a} \in \mathcal{A}, X \in \mathcal{X}, y \in \mathcal{Y}$:

$$\Pr\left(Y'_{A \leftarrow a}(U) = y | X = x, A = a\right) = \Pr\left(Y'_{A \leftarrow \breve{a}}(U) = y | X = x, A = a\right), \quad (3)$$

Figure 1: Causal graph in Example 4.1.

LCF implies that the subsequent consequence of ML decisions for a given individual in the factual world should be the same as that in the counterfactual world where the individual

belongs to other demographic groups. Note that CF may contradict LCF: even under counterfactually fair predictor, individuals in the factual and counterfactual worlds may end up with very different future statuses. We show this with an example below.

**Example 4.1.** Consider the causal graph in Figure 1 and the structural functions as follows:

$$
\begin{aligned}
X = f_X(U_1) = U_1, && Y = f_Y(U_2, X, A) = U_2 + X + A, \\
U_1' = r(U_1, \hat{Y}) = U_1 + \nabla_{U_1}\hat{Y}, && U_2' = r(U_2, \hat{Y}) = U_2 + \nabla_{U_2}\hat{Y}, \\
X' = f_X(U_1') = U_1', && Y' = f_Y(U_2', X', A) = U_2' + X' + A.
\end{aligned}
$$

Based on Kusner et al. (2017), a predictor that only uses $U_1$ and $U_2$ as input is counterfactually fair.[3] Therefore, $\hat{Y} = h(U_1, U_2)$ satisfies CF. Let $U_1$ and $U_2$ be uniformly distributed over $[-1, 1]$. Note that the response $r(U_1, \hat{Y})$ and $r(U_2, \hat{Y})$ imply that individuals make efforts to change feature vectors through changing the unobservable variables, which results in higher $\hat{Y}$ in the future. It is easy to see that a CF predictor $h(U_1, U_2) = U_1 + U_2$ minimizes the MSE loss $\mathbb{E}\{(Y - \hat{Y})^2\}$ if $A \in \{-1, 1\}$ and $\Pr\{A = 1\} = 0.5$. However, since $\nabla_{U_1}\hat{Y} = \nabla_{U_2}\hat{Y} = 1$, we have:

$$
\begin{aligned}
\Pr\left(Y_{A\leftarrow a}'(U) = y \mid X = x, A = a\right) = \delta(y - a - x - 2), \\
\Pr\left(Y_{A\leftarrow \check{a}}'(U) = y \mid X = x, A = a\right) = \delta(y - \check{a} - x - 2),
\end{aligned}
$$

where $\delta(y) = 1$ if $y = 0$ and $\delta(y) = 0$ otherwise. It shows that although the decisions in the factual and counterfactual worlds are the same, the future statuses $Y'$ are still different and Definition 4.1 does not hold.

Theorem 4.1 below identifies more general scenarios under which LCF can be violated with a CF predictor.

**Theorem 4.1** (**Violation of LCF under CF predictors**). *Consider a causal model $\mathcal{M} = (U, V, F)$ and individual response $r$ in the following form:*

$$
U' = r(U, \hat{Y}).
$$

*If the response $r$ is a function and the status $Y$ in factual and counterfactual worlds have different distributions, i.e.,*

$$
\Pr(Y_{A\leftarrow a}(U) = y \mid X = x, A = a) \neq \Pr(Y_{A\leftarrow \check{a}}(U) = y \mid X = x, A = a),
$$

*imposing any arbitrary model $\hat{Y}$ that satisfies CF will violate LCF, i.e.,*

$$
\Pr(Y_{A\leftarrow a}'(U) = y \mid X = x, A = a) \neq \Pr(Y_{A\leftarrow \check{a}}'(U) = y \mid X = x, A = a).
$$

## 5 Learning under LCF

This section introduces an algorithm for learning a predictor under LCF. In particular, we focus on a special case with the causal model and the individual response defined below.

Given sets of unobservable variables $U = \{U_1, ..., U_d, U_Y\}$ and observable variables $\{X_1, ..., X_d, A, Y\}$, we consider causal model with the following structural functions:

$$
X_i = f_i(U_i, A), \quad Y = f_Y(X_1, ..., X_d, U_Y), \tag{4}
$$

where $f_i$ is an invertible function[4], and $f_Y$ is invertible w.r.t. $U_Y$. After receiving the ML prediction $\hat{Y}$, the individual's future features $X'$ and status $Y'$ change accordingly. Specifically, we consider scenarios where individual unobservable variables $U$ change based on the following

$$
\begin{aligned}
U_i' = r_i(U_i, \hat{Y}) = U_i + \eta \nabla_{U_i}\hat{Y}, \quad \forall i \in \{1, ..., d\} \\
U_Y' = r_Y(U_Y, \hat{Y}) = U_Y + \eta \nabla_{U_Y}\hat{Y},
\end{aligned} \tag{5}
$$

---

[3]Note that $U_1$ and $U_2$ can be generated for each sample $(X, A)$. See Section 4.1 of (Kusner et al., 2017) for more details.

[4]Several works in causal inference also consider invertible structural function, e.g., *bijective causal models* introduced in Nasr-Esfahany et al. (2023).

and the future attributes $X_i'$ and status $Y'$ also change accordingly, i.e.,

$$X_i' = f_i(U_i', A),$$
$$Y' = f_Y(X_1', ..., X_d', U_Y'). \tag{6}$$

The above scenario implies that individuals respond to ML model by strategically moving features toward the **direction that increases their chances of receiving favorable decisions**, step size $\eta > 0$ controls the magnitude of data change and can be interpreted as the effort budget individuals have on changing their data. Note that this type of response has been widely studied in strategic classification literature (Rosenfeld et al., 2020; Hardt et al., 2016a). The above process with $d = 2$ is visualized in Figure 2.

Our goal is to train an ML model under LCF constraint. Before presenting our method, we first define the notion of counterfactual random variables.

**Definition 5.1** (Counterfactual random variable)**.** Let $x$ and $a$ be realizations of random variables $X$ and $A$, and $\breve{a} \neq a$. We say $\breve{X} := X_{A \leftarrow \breve{a}}(U)$ and $\breve{Y} := Y_{A \leftarrow \breve{a}}(U)$ are the counterfactual random variables associated with $(x, a)$ if $U$ follows the conditional distribution $\Pr\{U | X = x, A = a\}$ as given by the causal model $\mathcal{M}$. The realizations of $\breve{X}$, $\breve{Y}$ are denoted by $\breve{x}$ and $\breve{y}$.

The following theorem constructs a predictor $g$ that satisfies LCF, i.e., deploying the predictor $g$ in Theorem 5.1 ensures the future status $Y'$ is counterfactually fair.

**Theorem 5.1** (Predictor with perfect LCF)**.** *Consider causal model $\mathcal{M} = (U, V, F)$, where $U = \{U_X, U_Y\}$, $U_X = [U_1, U_2, ..., U_d]^{\mathrm{T}}$, $V = \{A, X, Y\}$, $X = [X_1, X_2, ..., X_d]^{\mathrm{T}}$, and the structural equations are given by,*

$$X = \alpha \odot U_X + \beta A, \quad Y = w^{\mathrm{T}} X + \gamma U_Y, \tag{7}$$

Figure 2: A causal graph and individual responses with two features $X_1, X_2$. The black arrows represent the connections described in structural functions. The red arrows represent the response process. The green dash arrows are the potential connection to prediction $\hat{Y}$.

*where $\alpha = [\alpha_1, \alpha_2, ..., \alpha_d]^{\mathrm{T}}$, $\beta = [\beta_1, \beta_2, ..., \beta_d]^{\mathrm{T}}$, $w = [w_1, w_2, .., w_d]^{\mathrm{T}}$, and $\odot$ denotes the element wise production. Then, the following predictor satisfies LCF,*

$$g(\breve{Y}, U) = p_1 \breve{Y}^2 + p_2 \breve{Y} + p_3 + h(U), \tag{8}$$

*where $p_1 = \frac{T}{2}$ with $T := \frac{1}{\eta(||w \odot \alpha||_2^2 + \gamma^2)}$, and $\breve{Y}$ is the counterfactual random variable associated with $Y$. Here, $p_2$ and $p_3$ and function $h(.)$ are arbitrary and can be trained to improve prediction performance.*

*Proof Sketch.* For any given $x, a$, we can find the conditional distribution of $U$. For a sample $u$ drawn from the distribution, we can compute $x, \breve{x}, y$ and $\breve{y}$. Then we have

$$g(\breve{y}, u) = p_1 \breve{y}^2 + p_2 \breve{y} + p_3 + h(u).$$

From this we can compute the gradient of $g$ w.r.t. $u_X$, $u_Y$. With response function $r$, we can get $u_X'$ and $u_Y'$. With the structural functions, we can know $y'$ and $\breve{y}'$. So, we have

$$|\breve{y}' - y'| = \left| \breve{y} - y + \frac{1}{T} \left( \frac{\partial g(y, u_X, u_Y)}{\partial y} - \frac{\partial g(\breve{y}, u_X, u_Y)}{\partial \breve{y}} \right) \right|.$$

Because we know that $\frac{\partial g(y, u_X, u_Y)}{\partial y} = 2p_1 y = Ty$, we have

$$|\breve{y}' - y'| = |y - \breve{y} + \breve{y} - y| = 0.$$

With law of total probability, we have LCF satisfied. □

The above theorem implies that $g$ should be constructed based on the counterfactual random variable $\check{Y}$ and $U$. Even though $U$ is unobserved, it can be obtained from the inverse of structural equations. Quantity $T$ in Theorem 5.1 depends on the step size $\eta$ in individual response, and parameters $\alpha, \gamma, w$ in structural functions. When $p_1 = \frac{T}{2}$, we can achieve perfect LCF.

It is worth noting that Definition 4.1 can be a very strong constraint and imposing $Y'_{A \leftarrow a}(U)$ and $Y'_{A \leftarrow \check{a}}(U)$ to have the same distribution may degrade the performance of the predictor significantly. To tackle this, we may consider a weaker version of LCF.

**Definition 5.2** (Relaxed LCF). We say Relaxed LCF holds if $\forall (a, \check{a}) \in \mathcal{A}^2, a \neq \check{a}, X \in \mathcal{X}, y \in \mathcal{Y}$, we have:

$$\Pr\left(\left\{\left|Y'_{A \leftarrow a}(U) - Y'_{A \leftarrow \check{a}}(U)\right| < \left|Y_{A \leftarrow a}(U) - Y_{A \leftarrow \check{a}}(U)\right|\right\} \Big| X = x, A = a\right) = 1. \tag{9}$$

Definition 5.2 implies that after individuals respond to ML model, the difference between the future status $Y'$ in factual and counterfactual worlds should be smaller than the difference between original status $Y$ in factual and counterfactual worlds. In other words, it means that the disparity between factual and counterfactual worlds must decrease over time. In Section 7, we empirically show that constraint in equation 9 is weaker than the constraint in equation 3 and can lead to a better prediction performance.

**Corollary 5.1** (Relaxed LCF with predictor in equation 8). *Consider the same causal model defined in Theorem 5.1 and the predictor defined in equation 8. Relaxed LCF holds if $p_1 \in (0, T)$.*

*Proof Sketch.* When $p_1 \in (0, T)$, from

$$|\check{y}' - y'| = \left|\check{y} - y + \frac{1}{T}\left(\frac{\partial g(y, u_X, u_Y)}{\partial y} - \frac{\partial g(\check{y}, u_X, u_Y)}{\partial \check{y}}\right)\right|,$$

we have

$$|\check{y}' - y'| = \left|y - \check{y} + \frac{2p_1}{T}(\check{y} - y)\right|.$$

Therefore $|\check{y}' - y'| < |\check{y} - y|$. With law of total probability, the Relaxed LCF is satisfied. $\qquad\square$

Apart from relaxing $p_1$ in predictor as shown in equation 8, we can also relax the form of the predictor to satisfy Relaxed LCF, as shown in Theorem 5.2.

**Theorem 5.2** (Predictor under Relaxed LCF). *Consider the same causal model defined in Theorem 5.1. A predictor $g(\check{Y}, U)$ satisfies Relaxed LCF if $g$ has the following three properties:*

*(i) $g(\check{y}, u)$ is strictly convex in $\check{y}$.*

*(ii) $g(\check{y}, u)$ can be expressed as $g(\check{y}, u) = g_1(\check{y}) + g_2(u)$.*

*(iii) The derivative of $g(\check{y}, u)$ w.r.t. $\check{y}$ is $K$-Lipschitz continuous in $\check{y}$ with $K < \frac{2}{\eta(\|w \odot \alpha\|_2^2 + \gamma^2)}$, i.e.,*

$$\left|\frac{\partial g(\check{y}_1, u)}{\partial \check{y}} - \frac{\partial g(\check{y}_2, u)}{\partial \check{y}}\right| \leq K \left|\check{y}_1 - \check{y}_2\right|.$$

*Proof Sketch.* When $g(\check{y}, u)$ satisfies property (ii), we can prove that

$$|\check{y}' - y'| = \left|\check{y} - y + \frac{1}{T}\left(\frac{\partial g(y, u_X, u_Y)}{\partial y} - \frac{\partial g(\check{y}, u_X, u_Y)}{\partial \check{y}}\right)\right|$$

still holds. Because of properties (i), we have

$$(\check{y} - y)\left(\frac{\partial g(y, u_X, u_Y)}{\partial y} - \frac{\partial g(\check{y}, u_X, u_Y)}{\partial y}\right) < 0.$$

Therefore, when $\left|\frac{\partial g(y, u_X, u_Y)}{\partial y} - \frac{\partial g(\check{y}, u_X, u_Y)}{\partial y}\right| < 2|\check{y} - y|$, $|y' - \check{y}'| < |y - \check{y}|$, which is guaranteed by property (iii). $\qquad\square$

Theorems 5.1 and 5.2 provide insights on designing algorithms to train a predictor with perfect or Relaxed LCF. Specifically, given training data $\mathcal{D} = \{(x^{(i)}, y^{(i)}, a^{(i)})\}_{i=1}^n$, we first estimate the structural equations. Then, we choose a parameterized predictor $g$ that satisfies the conditions in Theorem 5.1 or 5.2. An example is shown in Algorithm 1, which finds an optimal predictor in the form of $g(\check{y}, u) = p_1\check{y}^2 + p_2\check{y} + p_3 + h_\theta(u)$ under LCF, where $p_1 = \frac{1}{2\eta(||w\odot\alpha||_2^2+\gamma^2)}$, $\theta$ is the training parameter for function $h$, and $p_2, p_3$ are two other training parameters. Under Algorithm 1, we can find the optimal values for $p_2, p_3, \theta$ using training data $\mathcal{D}$. If we only want to satisfy Relaxed LCF (Definition 5.2), $p_1$ can be a training parameter with $0 < p_1 < T$.

---

**Algorithm 1** Training a predictor with perfect LCF

---

**Input:** Training data $\mathcal{D} = \{(x^{(i)}, y^{(i)}, a^{(i)})\}_{i=1}^n$, response parameter $\eta$.

1: Estimate the structural equations 7 using $\mathcal{D}$ to determine parameters $\alpha$, $\beta$, $w$, and $\gamma$.

2: For each data point $(x^{(i)}, y^{(i)}, a^{(i)})$, draw $m$ samples $\{u^{(i)[j]}\}_{j=1}^m$ from conditional distribution $\Pr\{U|X = x^{(i)}, A = a^{(i)}\}$ and generate counterfactual $\check{y}^{(i)[j]}$ associated with $u^{(i)[j]}$ based on structural equations 7.

3: Compute $p_1 \leftarrow \frac{1}{2\eta(||w\odot\alpha||_2^2+\gamma^2)}$.

4: Solve the following optimization problem,

$$\hat{p}_2, \hat{p}_3, \hat{\theta} = \arg\min_{p_2, p_3, \theta} \frac{1}{mn} \sum_{i=1}^n \sum_{j=1}^m l\left(g\left(\check{y}^{(i)[j]}, u^{(i)[j]}\right), y^{(i)}\right)$$

where

$$g\left(\check{y}^{(i)[j]}, u^{(i)[j]}\right) = p_1\left(\check{y}^{(i)[j]}\right)^2 + p_2\check{y}^{(i)[j]} + p_3 + h_\theta(u),$$

$\theta$ is a parameter for function $h$, and $l$ is a loss function.

**Output:** $\hat{p}_2, \hat{p}_3, \hat{\theta}$

---

It is worth noting that the results in Theorems 5.1 and 5.2 are for linear causal models. When the causal model is non-linear, it is hard to construct a model satisfying perfect LCF in Definition 4.1. Nonetheless, we can still show that it is possible to satisfy Relaxed LCF (Definition 5.2) for certain non-linear causal models. Theorem 5.3 below focuses on a special case when $X$ is not linearly dependent on $A$ and $U_X$ and it identifies the condition under which Relaxed LCF can be guaranteed.

**Theorem 5.3.** *Consider a bijective causal model $\mathcal{M} = (U, V, F)$, where $U$ is a scalar exogenous variable, $V = \{A, X, Y\}$, and the structural equations $X = f_X(A, U) \in \mathbb{R}^d, Y = f_Y(X, U) \in \mathbb{R}$ can be written in the form of*

$$Y = f(A, U) = f_Y(f_X(A, U), U) = \tilde{f}(U + u_0(A))$$

*for some function $\tilde{f}$, where $u_0(A)$ is an arbitrary function of $A$. Define function $\Gamma(s) = \tilde{f}(s)\frac{d\tilde{f}(s)}{ds}$ as the multiplication of $\tilde{f}(s)$ and its derivative. If $\tilde{f}(s)$ has the following properties:*

- *$\tilde{f}(s)$ is monotonic and strictly concave;*

- *If $\tilde{f}(s) \geq \tilde{f}(s')$ then $\tilde{f}(s)\tilde{f}'(s) \geq \tilde{f}(s')\tilde{f}'(s')$, $\forall s, s'$;*

- *$\Gamma(s) \geq 0$ and there exists constant $M > 0$ such that $\Gamma(s)$ is $M$-Lipschitz continuous.*

*Then, the following predictor satisfies Relaxed LCF,*

$$g(\check{Y}, U) = p_1\check{Y}^2 + p_2 + h(U),$$

*where $p_1 \in (0, \frac{1}{\eta M}]$, $p_2$ are learnable parameters, $\check{Y}$ is the counterfactual random variable associated with $Y$, and $h(U)$ can be an arbitrary monotonic function that is increasing (resp. decreasing) when $\tilde{f}(s)$ is increasing (resp. decreasing).*

*Proof Sketch.* For a sample $u$ drawn from the conditional distribution of $U$ given $X = x, A = a$, we can compute $y'$, $\check{y}'$, and get

$$\check{y} = \tilde{f}(\phi_1), \quad y = \tilde{f}(\phi_2), \quad \check{y}' = \tilde{f}(\phi_3), \quad y' = \tilde{f}(\phi_4).$$

The specific formulas of $\phi_1$, $\phi_2$, $\phi_3$ and $\phi_4$ can be seen in the full proof. With the mean value theorem, we know that

$$|y - \check{y}| = \tilde{f}'(c_1)|\phi_1 - \phi_2|,$$

where $c_1 \in (\phi_1, \phi_2)$, and

$$|y' - \check{y}'| = \tilde{f}'(c_2)|\phi_3 - \phi_4|,$$

where $c_2 \in (\phi_3, \phi_4)$. The three properties ensure that

$$\tilde{f}(c_1) > \tilde{f}(c_2), \quad |\phi_1 - \phi_2| \geq |\phi_3 - \phi_4|.$$

Therefore, $|\check{y}' - y'| < |\check{y} - y|$. With law of total probability, we have the Relaxed LCF satisfied. □

Theorems 5.2 and 5.3 show that designing a predictor under Relaxed LCF highly depends on the form of causal structure and structural equations. To wrap up this section, we would like to identify conditions under which Relaxed LCF holds in a causal graph that $X$ is determined by the product of $U_X$ and $A$.

**Theorem 5.4.** *Consider a non-linear causal model $\mathcal{M} = (U, V, F)$, where $U = \{U_X, U_Y\}$, $U_X = [U_1, U_2, ..., U_d]^{\mathrm{T}}, V = \{A, X, Y\}, X = [X_1, X_2, ..., X_d]^{\mathrm{T}}, A \in \{a_1, a_2\}$ is a binary sensitive attribute. Assume that the structural functions are given by,*

$$X = A \cdot (\alpha \odot U_X + \beta), \quad Y = w^{\mathrm{T}} X + \gamma U_Y, \tag{10}$$

*where $\alpha = [\alpha_1, \alpha_2, ..., \alpha_d]^{\mathrm{T}}$, $\beta = [\beta_1, \beta_2, ..., \beta_d]^{\mathrm{T}}$, and $\odot$ denotes the element wise production. A predictor $g(\check{Y})$ satisfies Relaxed LCF if $g$ and the causal model have the following three properties.*

    *(i) The value domain of $A$ satisfies $a_1 a_2 > 0$.*

    *(ii) $g(\check{y})$ is strictly convex.*

    *(iii) The derivate of $g(\check{y})$ is $K$-Lipschitz continuous with $K \leq \frac{2}{\eta(a_1 a_2 ||w \odot \alpha||_2^2 + \gamma^2)}$, i.e.,*

$$\left| \frac{\partial g(\check{y}_1)}{\partial \check{y}} - \frac{\partial g(\check{y}_2)}{\partial \check{y}} \right| < K |\check{y}_1 - \check{y}_2|.$$

*Proof Sketch.* For a sample $u$ drawn from the conditional distribution of $U$ given $X = x, A = a$, we can compute the $y'$ and $\check{y}'$ and get

$$|y' - \check{y}'| = |y - \check{y} + \Delta|.$$

The definition of $\Delta$ can be seen in the full proof. Because property (i) and property (ii),

$$(y - \check{y})\Delta < 0.$$

Property (iii) ensures that $|\Delta| < 2|y - \check{y}|$. Therefore, $|\check{y}' - y'| < |\check{y} - y|$. □

Although the structural equation associated with $Y$ is still linear in $X$ and $U_Y$, we emphasize that such a linear assumption has been very common in the literature due to the complex nature of strategic classification Zhang et al. (2022); Liu et al. (2020); Bechavod et al. (2022). For instance, Bechavod et al. (2022) assumed the actual status of individuals is $Y = \beta X$, a linear function of features $X$. Zhang et al. (2022) assumed that $X$ itself may be non-linear in some underlying traits of the individuals, but the relationship between $X$ and $P(Y = 1|X)$ is still linear. Indeed, due to the individual's strategic response, conducting the theoretical analysis accounting for such responses can be highly challenging. Nonetheless, it is worthwhile extending LCF to non-linear settings and we leave this for future works.

## 6 Path-dependent LCF

An extension of counterfactual fairness called path-dependent fairness has been introduced in Kusner et al. (2017). In this section, we also want to introduce an extension of LCF called path-dependent LCF. We will also modify Algorithm 1 to satisfy path-dependent LCF.

We start by introducing the notion of path-dependent counterfactuals. In a causal model associated with a causal graph $\mathcal{G}$, we denote $\mathcal{P}_{\mathcal{G}_A}$ as a set of unfair paths from sensitive attribute $A$ to $Y$. We define $X_{\mathcal{P}_{\mathcal{G}_A}^c}$ as the set of features that are not present in any of the unfair paths. Under observation $X = x, A = a$, we call $Y_{A \leftarrow \breve{a}, X_{\mathcal{P}_{\mathcal{G}_A}^c} \leftarrow x_{\mathcal{P}_{\mathcal{G}_A}^c}}(U)$ path-dependent counterfactual random variable for $Y$, and its distribution can be calculated as follows:

$$\Pr\{Y_{A \leftarrow \breve{a}, X_{\mathcal{P}_{\mathcal{G}_A}^c} \leftarrow x_{\mathcal{P}_{\mathcal{G}_A}^c}}(U) = y | X = x, A = a\} = \sum_u \Pr\{Y_{A \leftarrow \breve{a}, X_{\mathcal{P}_{\mathcal{G}_A}^c} \leftarrow x_{\mathcal{P}_{\mathcal{G}_A}^c}}(u) = y\} \Pr\{U = u | X = x, A = a\}.$$

For simplicity, we use $\breve{Y}_{PD}$ and $\breve{y}_{PD}$ to represent a path-dependent counterfactual and the corresponding realization. That is, $\breve{Y}_{PD} = Y_{A \leftarrow \breve{a}, X_{\mathcal{P}_{\mathcal{G}_A}^c} \leftarrow x_{\mathcal{P}_{\mathcal{G}_A}^c}}(U)$ where $U$ follows $\Pr\{U | X = x, A = a\}$. We consider the same kind of causal model described in Section 5, the future attributes $X'$ and outcome $Y'$ are determined by equation 5 and equation 6. We formally define the path-dependent LCF in the following definition.

**Definition 6.1.** We say an ML model satisfies path-dependent lookahead counterfactual fairness w.r.t. the unfair path set $\mathcal{P}_{\mathcal{G}_A}$ if the following holds $\forall a, \breve{a} \in \mathcal{A}, X \in \mathcal{X}, y \in \mathcal{Y}$:

$$\Pr\left(\hat{Y}'_{A \leftarrow a, X_{\mathcal{P}_{\mathcal{G}_A}^c} \leftarrow x_{\mathcal{P}_{\mathcal{G}_A}^c}}(U) = y \Big| X = x, A = a\right) = \Pr\left(\hat{Y}'_{A \leftarrow \breve{a}, X_{\mathcal{P}_{\mathcal{G}_A}^c} \leftarrow x_{\mathcal{P}_{\mathcal{G}_A}^c}}(U) = y \Big| X = x, A = a\right).$$

Then we have the following theorem.

**Theorem 6.1.** *Consider a causal model and structural equations defined in Theorem 5.1. If we denote the features on unfair path as $X_{\mathcal{P}_{\mathcal{G}_A}}$ and remaining features as $X_{\mathcal{P}_{\mathcal{G}_A}^c}$, we can re-write structural equations as*

$$
\begin{aligned}
X_{\mathcal{P}_{\mathcal{G}_A}} &= \alpha_{\mathcal{P}_{\mathcal{G}_A}} \odot U_{X_{\mathcal{P}_{\mathcal{G}_A}}} + \beta_{\mathcal{P}_{\mathcal{G}_A}} A, \\
X_{\mathcal{P}_{\mathcal{G}_A}^c} &= \alpha_{\mathcal{P}_{\mathcal{G}_A}^c} \odot U_{X_{\mathcal{P}_{\mathcal{G}_A}^c}} + \beta_{\mathcal{P}_{\mathcal{G}_A}^c} A, \\
Y &= w_{\mathcal{P}_{\mathcal{G}_A}}^{\mathrm{T}} X_{\mathcal{P}_{\mathcal{G}_A}} + w_{\mathcal{P}_{\mathcal{G}_A}^c}^{\mathrm{T}} X_{\mathcal{P}_{\mathcal{G}_A}^c} + \gamma U_Y
\end{aligned}
$$

*Then, the following predictor satisfies path-dependent LCF,*

$$g(\breve{Y}_{PD}, U) = p_1 \breve{Y}_{PD}^2 + p_2 \breve{Y}_{PD} + p_3 + h(U),$$

*where $p_1 = \frac{T}{2}$ with*

$$T := \frac{1}{\eta(||w_{\mathcal{P}_{\mathcal{G}_A}} \odot \alpha_{\mathcal{P}_{\mathcal{G}_A}}||_2^2 + ||w_{\mathcal{P}_{\mathcal{G}_A}^c} \odot \alpha_{\mathcal{P}_{\mathcal{G}_A}^c}||_2^2 + \gamma^2)},$$

*$p_2$ and $p_3$ are learnable parameters to improve prediction performance and $h$ is an arbitary function.*

*Proof Sketch.* Consider a sample $u$, we can compute $x_{\mathcal{P}_{\mathcal{G}_A}^c}, x_{\mathcal{P}_{\mathcal{G}_A}}, \breve{x}_{\mathcal{P}_{\mathcal{G}_A}}, y$ and $\breve{y}_{PD}$. Then we can get a similar equation about $|y' - \breve{y}'_{PD}|$ like Theorem 5.1. Then we can get $|y' - \breve{y}'_{PD}| = 0$ from the form of $g$. With law of total probability, we have the path-dependent LCF satisfied. □

## 7 Experiment

We conduct experiments on both synthetic and real data to validate the proposed method.

### 7.1 Synthetic Data

We generate the synthetic data based on the causal model described in Theorem 5.1, where we set $d = 10$ and generated 1000 data points. We assume $U_X$ and $U_Y$ follow the uniform distribution over $[0, 1]$ and the sensitive attribute $A \in \{0, 1\}$ is a Bernoulli random variable with $\Pr\{A = 0\} = 0.5$. Then, we generate $X$ and $Y$ using the structural functions described in Theorem 5.1.[5] Based on the causal model, the conditional distribution of $U_X$ and $U_Y$ given $X = x, A = a$ are as follows,

$$U_X | X = x, A = a \ \sim \ \delta\Big(\frac{x - \beta a}{\alpha}\Big), \quad U_Y | X = x, A = a \ \sim \ \text{Uniform}(0, 1). \tag{11}$$

**Baselines.**    We used two baselines for comparison: (i) **Unfair predictor (UF)** is a linear model without fairness constraint imposed. It takes feature $X$ as input and predicts $Y$. (ii) **Counterfactual fair predictor (CF)** only takes the unobservable variables $U$ as the input and was proposed by Kusner et al. (2017).

**Implementation Details.**    To find a predictor satisfying Definition 4.1, we train a predictor in the form of Eq. 8. In our experiment, $h(u)$ is a linear function. To train $g(\check{y}, u)$, we follows Algorithm 1 with $m = 100$. We split the dataset into the training/validation/test set at 60%/20%/20% ratio randomly and repeat the experiment 5 times. We use the validation set to find the optimal number of training epochs and the learning rate. Based on our observation, Adam optimization with a learning rate equal to $10^{-3}$ and 2000 epochs gives us the best performance.

**Metrics.**    We use three metrics to evaluate the methods. To evaluate the performance, we use the mean squared error (MSE). Given a dataset $\{x^{(i)}, a^{(i)}, y^{(i)}\}_{i=1}^n$, for each $x^{(i)}$ and $a^{(i)}$, we generate $m = 100$ values of $u^{(i)[j]}$ from the posterior distribution. MSE can be estimated as follows,[6]

$$\text{MSE} = \frac{1}{mn} \sum_{i=1}^n \sum_{j=1}^m \Big\| y^{(i)} - \hat{y}^{(i)[j]} \Big\|^2, \tag{12}$$

where $\hat{y}^{(i)[j]}$ is the prediction for data $(x^{(i)}, a^{(i)}, u^{(i)[j]})$. Note that for the UF baseline, the prediction does not depend on $u^{(i)[j]}$. Therefore, $\hat{y}^{(i)[j]}$ does not change by $j$ for the UF predictor. To evaluate fairness, we define a metric called average future causal effect (AFCE),

$$\text{AFCE} = \frac{1}{mn} \sum_{i=1}^n \sum_{j=1}^m \Big| y'^{(i)[j]} - \check{y}'^{(i)[j]} \Big|.$$

It is the average difference between the factual and counterfactual future outcomes. To compare $|Y - \check{Y}|$ with $|Y' - \check{Y}'|$ under different algorithms, we use the unfairness improvement ratio (UIR) defined below. The larger UIR implies a higher improvement in disparity.

$$\text{UIR} = \left(1 - \frac{\sum_{i=1}^n \sum_{j=1}^m |y'^{(i)[j]} - \check{y}'^{(i)[j]}|}{\sum_{i=1}^n \sum_{j=1}^m |y^{(i)[j]} - \check{y}^{(i)[j]}|}\right) \times 100\%.$$

**Results.**    Table 1 illustrates the results when we set $\eta = 10$ and $p_1 = \frac{T}{2}$. The results show that our method can achieve perfect LCF with $p_1 = \frac{T}{2}$. Note that in our experiment, the range of $Y$ is $[0, 3.73]$, and our method and UF can achieve similar MSE. Moreover, our method achieves better performance than the CF method because $\check{Y}$ includes useful predictive information and using it in our predictor can improve performance and decrease the disparity simultaneously. Because both CF and UF do not take into account future outcome $Y'$, $|Y' - \check{Y}'|$ is similar to $|Y - \check{Y}|$, leading to UIR = 0. Based on Corollary 5.1, the value of $p_1$ can impact the strength of fairness. We examine the tradeoff between accuracy and fairness by changing

---

[5]The exact values for parameters $\alpha$, $\beta$, $w$ and $\gamma$ can be found in the Appendix B.
[6]Check Section 4.1 of Kusner et al. (2017) for details on why equation 12 is an empirical estimate of MSE.

Table 1: Results on Synthetic Data: comparison with two baselines, unfair predictor (UF) and counterfactual fair predictor (CF), in terms of accuracy (MSE) and lookahead counterfactual fairness (AFCE, UIR).

| Method | MSE | AFCE | UIR |
|---|---|---|---|
| UF | $0.036 \pm 0.003$ | $1.296 \pm 0.000$ | $0\% \pm 0$ |
| CF | $0.520 \pm 0.045$ | $1.296 \pm 0.000$ | $0\% \pm 0$ |
| Ours ($p_1 = T/2$) | $0.064 \pm 0.001$ | $0.000 \pm 0.0016$ | $100\% \pm 0$ |

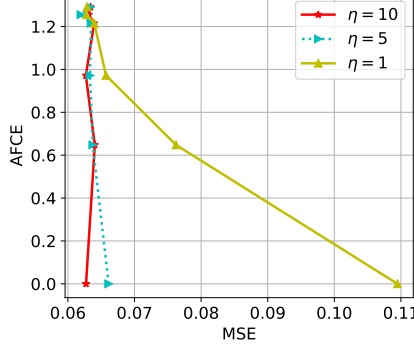

(a) Accuracy-fairness trade-off of predictor Eq. 8 on synthetic data: we vary $p_1$ from $\frac{T}{512}$ to $\frac{T}{2}$ under different $\eta$. When $p_1 = \frac{T}{2}$, we attain perfect LCF.

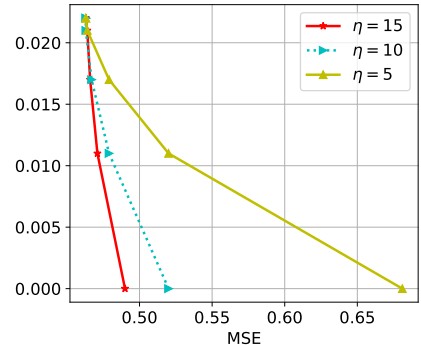

(b) Accuracy-fairness trade-off on the law school dataset: we vary $p_1$ from $\frac{T}{512}$ to $\frac{T}{2}$ under different $\eta$. When $p_1 = \frac{T}{2}$, we attain perfect LCF.

the value of $p_1$ from $\frac{T}{512}$ to $\frac{T}{2}$ under different $\eta$. Figure 3a shows the MSE as a function of AFCE. The results show that when $\eta = 1$ we can easily control the accuracy-fairness trade-off in our algorithm by adjusting $p_1$. When $\eta$ becomes large, we can get a high LCF improvement while maintaining a low MSE. To show how our method impacts a specific individual, we choose the first data point in our test dataset and plot the distribution of factual future status $Y'$ and counterfactual future status $\check{Y}'$ for this specific data point under different methods. Figure 4 illustrates such distributions. It can be seen in the most left plot that there is an obvious gap between factual $Y$ and counterfactual $\check{Y}$. Both UF and CF can not decrease this gap for future outcome $Y'$. However, with our method, we can observe that the distributions of $Y'$ and $\check{Y}'$ become closer to each other. When $p_1 = \frac{T}{2}$ (the most right plot in Figure 4), the two distributions become the same in the factual and counterfactual worlds.

## 7.2 Real Data: The Law School Success Dataset

We further measure the performance of our proposed method using the Law School Admission Dataset Wightman (1998). In this experiment, the objective is to forecast the first-year average grades (FYA) of students in law school using their undergraduate GPA and LSAT scores.

**Dataset.** The dataset consists of 21,791 records. Each record is characterized by 4 attributes: Sex ($S$), Race ($R$), UGPA ($G$), LSAT ($L$), and FYA ($F$). Both Sex and Race are categorical in nature. The Sex attribute can be either male or female, while Race can be Amerindian, Asian, Black, Hispanic, Mexican, Puerto Rican, White, or other. The UGPA is a continuous variable ranging from 0 to 4. LSAT is an integer-based attribute with a range of $[0, 60]$. FYA, which is the target variable for prediction, is a real number ranging from $-4$ to 4 (it has been normalized). In this study, we consider $S$ as the sensitive attribute, while $R, G$, and $L$ are treated as features.

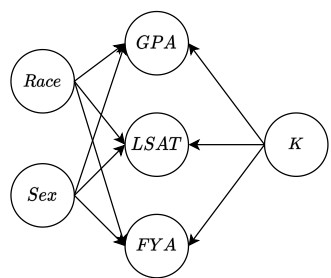

Figure 5: Causal model for the Law School Dataset.

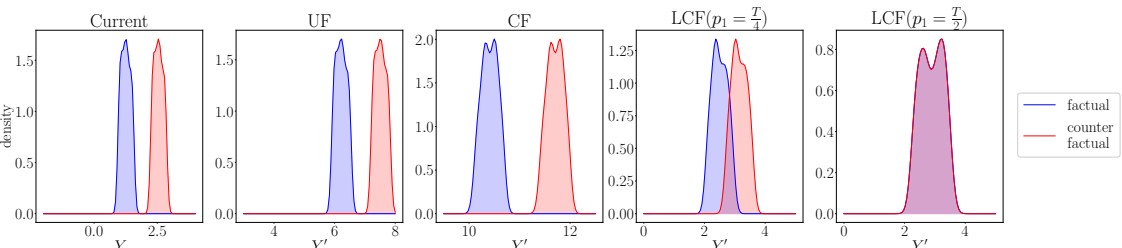

Figure 4: Density plot for $Y'$ and $\check{Y}'$ in synthetic data. For a chosen data point, we sampled a batch of $U$ under the conditional distribution of it and plot the distribution of $Y'$ and $\check{Y}'$.

**Causal Model.** We adopt the causal model as presented in Kusner et al. (2017), which can be visualized in Figure 5. In this causal graph, $K$ represents an unobserved variable, which can be interpreted as *knowledge*. Thus, the model suggests that students' grades (UGPA, LSAT, FYA) are influenced by their sex, race, and underlying knowledge. We assume that the prior distribution for $K$ follows a normal distribution, denoted as $\mathcal{N}(0,1)$. We adopt the same structural equations as Kusner et al. (2017):

$$
\begin{aligned}
G &= \mathcal{N}(w_G^K K + w_G^R R + w_G^S S + b_G, \sigma_G), \\
L &= \text{Poisson}\left(\exp\left\{w_L^K K + w_L^R R + w_L^S S + b_L\right\}\right), \\
F &= \mathcal{N}(w_F^K K + w_F^R R + w_F^S S, 1).
\end{aligned}
$$

**Implementation Details.** Note that race is an immutable characteristic. Therefore, we assume that the individuals only adjust their knowledge $K$ in response to the prediction model $\hat{Y}$. That is $K' = K + \eta \nabla_K \hat{Y}$. In contrast to synthetic data, the parameters of structural equations are unknown, and we have to use the training dataset to estimate them. Following the approach of Kusner et al. (2017), we assume that $G$ and $F$ adhere to Gaussian distributions centered at $w_G^K K + w_G^R R + w_G^S S + b_G$ and $w_F^K K + w_F^R R + w_F^S S$, respectively. Note that $L$ is an integer, and it follows a Poisson distribution with the parameter $\exp\{w_L^K K + w_L^R R + w_L^S S + b_L\}$. Using the Markov chain Monte Carlo (MCMC) method Geyer (1992), we can estimate the parameters and the conditional distribution of $K$ given $(R, S, G, L)$. For each given data, we sampled $m = 500$ different $k$'s from this conditional distribution. We partitioned the data into training, validation, and test sets with $60\%/20\%/20\%$ ratio.

Table 2: Results on Law School Data: comparison with two baselines, unfair predictor (UF) and counterfactual fair predictor (CF), in terms of accuracy (MSE) and lookahead counterfactual fairness (AFCE, UIR).

| Method | MSE | AFCE | UIR |
|---|---|---|---|
| UF | $0.393 \pm 0.046$ | $0.026 \pm 0.003$ | $0\% \pm 0$ |
| CF | $0.496 \pm 0.051$ | $0.026 \pm 0.003$ | $0\% \pm 0$ |
| Ours $(p_1 = T/4)$ | $0.493 \pm 0.049$ | $0.013 \pm 0.002$ | $50\% \pm 0$ |
| Ours $(p_1 = T/2)$ | $0.529 \pm 0.049$ | $0.000 \pm 0.000$ | $100\% \pm 0$ |

**Results.** Table 2 illustrates the results with $\eta = 10$ and $p_1 = \frac{T}{4}$ and $p_1 = \frac{T}{2}$ where $T = 1/(w_K^F)^2$. The results show that our method achieves a similar MSE as the CF predictor. However, it can improve AFCE significantly compared to the baselines. Figure 6 shows the distribution of $Y$ and $Y'$ for the first data point in the test set in the factual and counterfactual worlds. Under the UF and CF predictor, the disparity between factual and factual $Y'$ remains similar to the disparity between factual and counterfactual $Y$. On the other hand, the disparity between factual and counterfactual $Y'$ under our algorithms gets better for both $p_1 = T/2$ and $p_1 = T/4$. Figure 3b demonstrates that for the law school dataset, the trade-off between MSE and AFCE can be adjusted by changing hyperparameter $p_1$. Figure 6 show the factual and counterfactual distributions in real data experiment. It can be seen that our method is the only way that can decrease the gap between $Y'$ and $\check{Y}'$ in an obvious way.

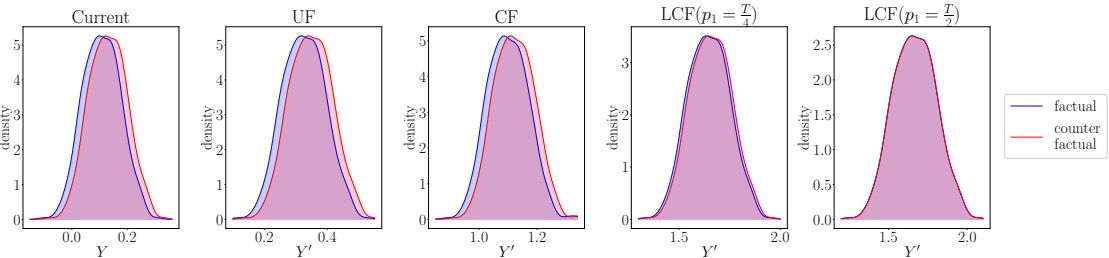

Figure 6: Density plot for $F'$ and $\check{F}'$ in law school data. For a chosen data point, we sampled $K$ from the conditional distribution of $K$ and plot the distribution of $F'$ and $\check{F}'$.

# 8 Conclusion

This work studied the impact of ML decisions on individuals' future status using a counterfactual inference framework. We observed that imposing the CF predictor may not decrease the group disparity in individuals' future status. We thus introduced the lookahead counterfactual fairness (LCF) notion, which takes into account the downstream effects of ML models and requires the individual future status to be counterfactually fair. We proposed a method to train an ML model under LCF and evaluated the method through empirical studies on synthetic and real data.

## Acknowledgements

This material is based upon work supported by the U.S. National Science Foundation under award IIS-2202699, IIS-2416895, IIS-2301599, and CMMI-2301601, and by OSU President's Research Excellence Accelerator Grant, and grants from the Ohio State University's Translational Data Analytics Institute and College of Engineering Strategic Research Initiative.

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

# A  Proofs

## A.1  Proof of Theorem 5.1 and Theorem 5.2

*Proof.* For any given $x, a$, we can find the conditional distribution $U_X | X = x, A = a$ and $U_Y | X = x, A = a$ based on causal model $\mathcal{M}$. Consider sample $u = [u_X, u_Y]$ drawn from this conditional distribution. For this sample, we have,

$$\check{x} = \alpha \odot u_X + \beta \check{a},$$
$$\check{y} = w^{\mathrm{T}} \check{x} + \gamma u_Y.$$

Since $\check{y}$ is also a function of $u_X, u_Y$, utilizing that

$$\frac{\partial \check{y}}{\partial u_X} = w \odot \alpha, \quad \frac{\partial \check{y}}{\partial u_Y} = \gamma,$$

the gradient of $g(\check{y}, u_X, u_Y)$ w.r.t. $u_X, u_Y$ are

$$\nabla_{u_X} g = \frac{\partial g(\check{y}, u_X, u_Y)}{\partial u_X} + \frac{\partial g(\check{y}, u_X, u_Y)}{\partial \check{y}} w \odot \alpha,$$
$$\nabla_{u_Y} g = \frac{\partial g(\check{y}, u_X, u_Y)}{\partial u_Y} + \frac{\partial g(\check{y}, u_X, u_Y)}{\partial \check{y}} \gamma.$$

The response function $r$ is defined as

$$u_X^{'} = u_X + \nabla_{u_X} g, \quad u_Y^{'} = u_Y + \nabla_{u_Y} g.$$

$y'$ can be calculated using response $r$ as follows,

$$y' = y + \eta w^{\mathrm{T}} \left( \alpha \odot \frac{\partial g(\check{y}, u_X, u_Y)}{\partial u_X} \right) + \eta \, || w \odot \alpha \, ||_2^2 \frac{\partial g(\check{y}, u_X, u_Y)}{\partial \check{y}} + \eta \gamma \frac{\partial g(\check{y}, u_X, u_Y)}{\partial u_Y} + \eta \gamma^2 \frac{\partial g(\check{y}, u_X, u_Y)}{\partial \check{y}}. \tag{13}$$

Similarly, we can calculate counterfactual value $\check{y}'$ as follows,

$$\check{y}' = \check{y} + \eta w^{\mathrm{T}} \left( \alpha \odot \frac{\partial g(y, u_X, u_Y)}{\partial u_X} \right) + \eta \, || w \odot \alpha | \, |_2^2 \frac{\partial g(y, u_X, u_Y)}{\partial y} + \eta \gamma \frac{\partial g(y, u_X, u_Y)}{\partial u_Y} + \eta \gamma^2 \frac{\partial g(y, u_X, u_Y)}{\partial y}. \tag{14}$$

Note that the following hold for $g$,

$$\frac{\partial g(\check{y}, u_X, u_Y)}{\partial u_X} = \frac{\partial g(y, u_X, u_Y)}{\partial u_X}, \tag{15}$$
$$\frac{\partial g(\check{y}, u_X, u_Y)}{\partial u_Y} = \frac{\partial g(y, u_X, u_Y)}{\partial u_Y}. \tag{16}$$

Thus,

$$|\check{y}' - y'| = \left| \check{y} - y + \eta \left( ||w \odot \alpha||_2^2 + \gamma^2 \right) \left( \frac{\partial g(y, u_X, u_Y)}{\partial y} - \frac{\partial g(\check{y}, u_X, u_Y)}{\partial \check{y}} \right) \right|. \tag{17}$$

Given above equation, now we can prove Theorem 5.1, Corollary 5.1, and Theorem 5.2,

- For $g$ in Theorem 5.1 and Corollary 5.1, we have,

$$g(\check{y}, u_X, u_Y) = p_1 \check{y}^2 + p_2 \check{y} + p_3 + h(u). \tag{18}$$

The partial derivative of $g$ can be computed as

$$\frac{\partial g(\check{y}, u_X, u_Y)}{\partial \check{y}} = 2 p_1 \check{y} + p_2. \tag{19}$$

We denote $T = \eta \left( ||w \odot \alpha||_2^2 + \gamma^2 \right)$. From the theorem, we know that $p_1 = \frac{T}{2}$. Therefore, we have

$$|y' - \check{y}'| = |\check{y} - y + \frac{1}{T}T(y - \check{y})| = 0.$$

Since, for any realization of $u$, the above equation holds, we can conclude that the following holds,

$$\Pr(\hat{Y}_{A \leftarrow a}(U) = y | X = x, A = a) = \Pr(\hat{Y}_{A \leftarrow \check{a}}(U) = y | X = x, A = a)$$

When $p_1 \in (0, T)$, we have that

$$|y' - \check{y}'| = |\check{y} - y + \frac{2}{T}p_1(y - \check{y})| = 0.$$

Because

$$(\check{y} - y) \left( \frac{2p_1}{T}(y - \check{y}) \right) < 0,$$

and

$$\left| \frac{2p_1}{T}(y - \check{y}) \right| < 2|\check{y} - y|,$$

we have $|y' - \check{y}'| < |y - \check{y}|$. With law of total probability, we have

$$\Pr(\{|Y'_{A \leftarrow a}(U) - Y'_{A \leftarrow \check{a}}(U)| < |Y_{A \leftarrow a}(U) - Y_{A \leftarrow \check{a}}(U)|\} | X = x, A = a) = 1.$$

- For $g$ in Theorem 5.2, since $g(\check{y}, u_X, u_Y)$ is strictly convex in $\check{y}$, we have,

$$(\check{y} - y) \left( \frac{\partial g(y, u_X, u_Y)}{\partial y} - \frac{\partial g(\check{y}, u_X, u_Y)}{\partial \check{y}} \right) > 0.$$

Note that derivative of $g(\check{y}, u_X, u_Y)$ with respect to $\check{y}$ is $K$-Lipschitz continuous in $\check{y}$,

$$\left| \frac{\partial g(y, u_X, u_Y)}{\partial y} - \frac{\partial g(\check{y}, u_X, u_Y)}{\partial \check{y}} \right| < \frac{2|y - \check{y}|}{\eta(||w \odot \alpha||_2^2 + \gamma^2)},$$

we have that

$$\left| \eta \left( ||w \odot \alpha||_2^2 + \gamma^2 \right) \left( \frac{\partial g(y, u_X, u_Y)}{\partial y} - \frac{\partial g(\check{y}, u_X, u_Y)}{\partial \check{y}} \right) \right| < 2|\check{y} - y|.$$

Therefore,

$$|y' - \check{y}'| < |y - \check{y}|$$

So we have

$$\Pr(\{|Y'_{A \leftarrow a}(U) - Y'_{A \leftarrow \check{a}}(U)| < |Y_{A \leftarrow a}(U) - Y_{A \leftarrow \check{a}}(U)|\} | X = x, A = a) = 1$$

$\square$

## A.2   Theorem 5.2 for non-binary $A$

Let $\{a\} \cup \{\check{a}^{[1]}, \check{a}^{[2]}, ..., \check{a}^{[m]}\}$ be a set of all possible values for $A$. Let $\check{Y}^{[j]}$ be the counterfactual random variable associated with $\check{a}^{[j]}$ given observation $X = x$ and $A = a$. Then, $g \left( \frac{1}{m}(\check{Y}^{[1]} + \cdots \check{Y}^{[m]}), U \right)$ satisfies LCF, where $g$ satisfies the properties in Theorem 5.2.

*Proof.* For any given $x, a$, we assume the set of counterfactual $a$ is $\{\check{a}^{[1]}, \check{a}^{[2]}, ..., \check{a}^{[m]}\}$. Consider a sample $u = [u_X, u_Y]$ drawn from the condition distribution of $U_X|X = x, A = a$ and $U_Y|X = x, A = a$, with a predictor $g\left(\frac{1}{m}(\check{y}^{[1]} + \cdots \check{y}^{[m]}), u\right)$, use the same way in A.1, we can get

$$\left|\check{y}'^{[j]} - y'\right| = \left|\check{y}^{[j]} - y + \eta(||w \odot \alpha||^2 + \gamma^2)\left(\frac{\partial g(\check{y}^{[1]} + \cdots \check{y}^{[m]}, u)}{\partial \check{y}^{[1]} + \cdots \check{y}^{[m]}} - \frac{\partial g(y + \check{y}^{[1]} + \cdots + \check{y}^{[j-1]} + \check{y}^{[j+1]} \cdots \check{y}^{[m]}, u)}{\partial y + \check{y}^{[1]} + \cdots \check{y}^{[j-1]} + \check{y}^{[j+1]} \cdots \check{y}^{[m]}}\right)\right|,$$

with $j \in \{1, ..., m\}$. When $y > \check{y}^{[j]}$, we have

$$\check{y}^{[1]} + \cdots \check{y}^{[m]} < y + \check{y}^{[1]} + \cdots \check{y}^{[j-1]} + \check{y}^{[j+1]} \cdots \check{y}^{[m]}$$

and when $y < \check{y}^{[j]}$,

$$\check{y}^{[1]} + \cdots \check{y}^{[m]} > y + \check{y}^{[1]} + \cdots \check{y}^{[j-1]} + \check{y}^{[j+1]} \cdots \check{y}^{[m]}$$

Because $g$ is strictly convex and Lipschitz continuous, we have

$$(\check{y}^{[j]} - y)\left(\frac{\partial g(\check{y}^{[1]} + \cdots \check{y}^{[m]}, u)}{\partial \check{y}^{[1]} + \cdots \check{y}^{[m]}} - \frac{\partial g(y + \check{y}^{[1]} + \cdots + \check{y}^{[j-1]} + \check{y}^{[j+1]} \cdots \check{y}^{[m]}, u)}{\partial y + \check{y}^{[1]} + \cdots \check{y}^{[j-1]} + \check{y}^{[j+1]} \cdots \check{y}^{[m]}}\right) < 0,$$

and

$$\left|\eta(||w \odot \alpha||^2 + \gamma^2)\left(\frac{\partial g(\check{y}^{[1]} + \cdots \check{y}^{[m]}, u)}{\partial \check{y}^{[1]} + \cdots \check{y}^{[m]}} - \frac{\partial g(y + \check{y}^{[1]} + \cdots + \check{y}^{[j-1]} + \check{y}^{[j+1]} \cdots \check{y}^{[m]}, u)}{\partial y + \check{y}^{[1]} + \cdots \check{y}^{[j-1]} + \check{y}^{[j+1]} \cdots \check{y}^{[m]}}\right)\right| < 2|\check{y}^{[j]} - y|.$$

Therefore,

$$|\check{y}'^{[j]} - y'| < |\check{y}^{[j]} - y|$$

So we proved that, for any $j \in \{1, 2, ..., m\}$

$$\Pr(\{|Y'_{A \leftarrow a}(U) - Y'_{A \leftarrow \check{a}^{[j]}}(U)| < |Y_{A \leftarrow a}(U) - Y_{A \leftarrow \check{a}^{[j]}}(U)|\}|X = x, A = a) = 1$$

$\square$

## A.3 Proof of Theorem 5.3

*Proof.* We start from the case when $\tilde{f}$ is increasing. For any given $x, a$, we can find the conditional distribution $U|X = x, A = a$ based on the causal model $\mathcal{M}$. Consider a sample $u$ drawn from this conditional distribution. For this sample, we have

$$y = \tilde{f}(u + u_0(a)), \quad \check{y} = \tilde{f}(u + u_0(\check{a})).$$

So, the gradient of $g(\check{y}, u) = p_1\check{y}^2 + p_2 + h(u)$ w.r.t. $u$ is

$$\frac{dg(\check{y}, u)}{du} = 2p_1\check{y}\frac{d\check{y}}{du} + \frac{dh(u)}{du}.$$

Therefore, $y'$ can be calculated using response $r$ as follows,

$$y' = \tilde{f}\left(u + u_0(a) + 2\eta p_1\check{y}\frac{d\check{y}}{du} + \eta\frac{dh(u)}{du}\right).$$

Similarly, we have the future counterfactual status as

$$\check{y}' = \tilde{f}\left(u + u_0(\check{a}) + 2\eta p_1 y\frac{dy}{du} + \eta\frac{dh(u)}{du}\right).$$

When $y = \check{y}$, it is obvious that $y' = \check{y}'$ since $\check{y}\frac{d\check{y}}{du} = y\frac{dy}{du}$. When $y > \check{y}$, because $\tilde{f}$ is increasing, we have

$$u_0(a) > u_0(\check{a}).$$

Because $h(U)$ is increasing, we have $\frac{dh(u)}{du} \geq 0$ and

$$2\eta p_1 \check{y} \frac{d\check{y}}{du} + \eta \frac{dh(u)}{du} \geq 0, \quad 2\eta p_1 y \frac{dy}{du} + \eta \frac{dh(u)}{du} \geq 0.$$

We further denote

$$\phi_1 = u + u_0(\check{a})$$
$$\phi_2 = u + u_0(a)$$
$$\phi_3 = u + u_0(\check{a}) + 2\eta p_1 y \frac{dy}{du} + \eta \frac{dh(u)}{du}$$
$$\phi_4 = u + u_0(a) + 2\eta p_1 \check{y} \frac{d\check{y}}{du} + \eta \frac{dh(u)}{du}$$

We already have

$$\phi_1 < \phi_2, \quad \phi_1 \leq \phi_3, \quad \phi_2 \leq \phi_4.$$

There are three cases for the relationship between $\phi_1$, $\phi_2$, $\phi_3$ and $\phi_4$.

**Case 1:** $\phi_1 < \phi_2 \leq \phi_3 \leq \phi_4$. In this case, from the mean value theorem,

$$|y - \check{y}| = \left|\tilde{f}(\phi_2) - \tilde{f}(\phi_1)\right| = \left|\tilde{f}'(c_1)\right| |\phi_2 - \phi_1|,$$

where $c_1 \in (\phi_1, \phi_2)$, and

$$|y' - \check{y}'| = \left|\tilde{f}(\phi_4) - \tilde{f}(\phi_3)\right| = \left|\tilde{f}'(c_2)\right| |\phi_4 - \phi_3|,$$

where $c_2 \in (\phi_3, \phi_4)$. Because

$$|\phi_4 - \phi_3| = (u_0(a) - u_0(\check{a})) + 2\eta p_1 \left(\check{y} \frac{d\check{y}}{du} - y \frac{dy}{du}\right),$$

and

$$\check{y} \frac{d\check{y}}{du} \leq y \frac{dy}{du},$$

we have

$$|\phi_4 - \phi_3| \leq u_0(a) - u_0(\check{a}) = |\phi_2 - \phi_1|.$$

Because $\tilde{f}$ is strictly concave, $\tilde{f}'$ is strictly decreasing, and we have,

$$\left|\tilde{f}'(c_1)\right| > \left|\tilde{f}'(c_2)\right|.$$

Therefore,

$$|y' - \check{y}'| < |y - y'|.$$

**Case 2:** $\phi_1 \leq \phi_3 < \phi_2 \leq \phi_4$. In this case,

$$|y - \check{y}| = \tilde{f}(\phi_2) - \tilde{f}(\phi_1) = \tilde{f}(\phi_2) - \tilde{f}(\phi_3) + \tilde{f}(\phi_3) - \tilde{f}(\phi_1),$$
$$|y' - \check{y}'| = \tilde{f}(\phi_4) - \tilde{f}(\phi_3) = \tilde{f}(\phi_4) - \tilde{f}(\phi_2) + \tilde{f}(\phi_2) - \tilde{f}(\phi_3).$$

From the mean value theorem,

$$\left|\tilde{f}(\phi_3) - \tilde{f}(\phi_1)\right| = \left|\tilde{f}'(c_1)\right| |\phi_3 - \phi_1|,$$

where $c_1 \in (\phi_1, \phi_3)$ and

$$\left| \tilde{f}(\phi_4) - \tilde{f}(\phi_2) \right| = \left| \tilde{f}'(c_2) \right| |\phi_4 - \phi_2|,$$

where $c_2 \in (\phi_2, \phi_4)$. Because $\tilde{f}'$ is decreasing, we have

$$\left| \tilde{f}'(c_1) \right| > \left| \tilde{f}'(c_2) \right|.$$

Because

$$|\phi_4 - \phi_2| = 2\eta p_1 \check{y} \frac{d\check{y}}{du} + \eta \frac{dh(u)}{du}$$

$$|\phi_3 - \phi_1| = 2\eta p_1 y \frac{dy}{du} + \eta \frac{dh(u)}{du},$$

we have that $|\phi_3 - \phi_1| \le |\phi_4 - \phi_2|$. Therefore, $|y' - \check{y}'| < |y - y'|$.

**Case 3:** $\phi_1 < \phi_2 \le \phi_4 \le \phi_3$. In this case, we have

$$|y' - \check{y}'| = \tilde{f}(\phi_3) - \tilde{f}(\phi_4).$$

From the mean value theorem,

$$|y - \check{y}| = \left| \tilde{f}'(c_1) \right| |\phi_2 - \phi_1|,$$

where $c_1 \in (\phi_1, \phi_2)$, and

$$|y' - \check{y}'| = \left| \tilde{f}'(c_2) \right| |\phi_3 - \phi_4|,$$

where $c_2 \in (\phi_4, \phi_3)$. Because

$$|\phi_2 - \phi_1| = u_0(a) - u_0(\check{a});$$

$$|\phi_3 - \phi_4| = (u_0(\check{a}) - u_0(a)) + 2\eta p_1 \left( y \frac{dy}{du} - \check{y} \frac{d\check{y}}{du} \right),$$

and

$$\left| \frac{y \frac{dy}{du} - \check{y} \frac{d\check{y}}{du}}{(u + u_0(a)) - (u + u_0(\check{a}))} \right| \le M,$$

when $p_1 \le \frac{1}{\eta M}$,

$$|\phi_3 - \phi_4| \le (u_0(\check{a}) - u_0(a)) + 2(u_0(a) - u_0(\check{a})) \le u_0(a) - u_0(\check{a}) = |\phi_2 - \phi_1|$$

Because $\tilde{f}'(c_1) > \tilde{f}'(c_2)$, we have $|y' - \check{y}'| < |y - \check{y}|$.

In conclusion, we prove that $|y' - \check{y}'| < |y - \check{y}|$ for every sample $u$. With law of total probability, we have

$$\Pr \left( \left\{ |Y'_{A \leftarrow a}(U) - Y'_{A \leftarrow \check{a}}(U)| < |Y_{A \leftarrow a}(U) - Y_{A \leftarrow \check{a}}(U)| \right\} \Big| X = x, A = a \right) = 1.$$

For the case when $\tilde{f}$ is decreasing, we can consider $-Y = -\tilde{f}(U + u_0(A))$, then we have

$$\Pr \left( \left\{ |-Y'_{A \leftarrow a}(U) - (-Y'_{A \leftarrow \check{a}}(U))| < |-Y_{A \leftarrow a}(U) - (-Y_{A \leftarrow \check{a}}(U))| \right\} \Big| X = x, A = a \right) = 1,$$

which is to say

$$\Pr \left( \left\{ |Y'_{A \leftarrow a}(U) - Y'_{A \leftarrow \check{a}}(U)| < |Y_{A \leftarrow a}(U) - Y_{A \leftarrow \check{a}}(U)| \right\} \Big| X = x, A = a \right) = 1.$$

$\square$

### A.4 Proof of Theorem 5.4

*Proof.* From the causal functions defined in Theorem 5.4, given any $x, a$, we can find the conditional distribution $U_X|X = x, A = a$ and $U_Y|X = x, A = a$. Similar to the proof of Theorem 5.2, we have

$$\check{x} = \check{a}(\alpha \odot u_X + \beta),$$
$$\check{y} = w^{\mathrm{T}}\check{x} + \gamma u_Y.$$

Because

$$\frac{\partial \check{y}}{\partial u_X} = w \odot \check{a}\alpha, \quad \frac{\partial \check{y}}{\partial u_Y} = \gamma,$$

the gradient of $g(\check{y})$ w.r.t $u_X, u_Y$ are

$$\nabla_{u_X} g = \frac{\partial g(\check{y})}{\partial \check{y}} \check{a}w \odot \alpha,$$

$$\nabla_{u_Y} g = \frac{\partial g(\check{y})}{\partial \check{y}} \gamma.$$

The response function $r$ is defined as

$$u_X^{'} = u_X + \nabla_{u_X} g, \quad u_Y^{'} = u_Y + \nabla_{u_Y} g.$$

$y'$ can be calculated using the response $r$ as follows,

$$y' = y + \eta \left( a\check{a}||w \odot \alpha||^2 + \gamma^2 \right) \frac{\partial g(\check{y})}{\partial \check{y}}.$$

In the counterfactual world,

$$\check{y}' = \check{y} + \eta(\check{a}a||w \odot \alpha||^2 + \gamma^2) \frac{\partial g(y)}{\partial y}.$$

So we have,

$$|y' - \check{y}'| = \left| y - \check{y} + \eta(a\check{a}||w \odot \alpha||^2 + \gamma^2)(\frac{\partial g(\check{y})}{\partial \check{y}} - \frac{\partial g(y)}{\partial y}) \right|.$$

We denote that $\Delta = \eta(a\check{a}||w \odot \alpha||^2 + \gamma^2)(\frac{\partial g(\check{y})}{\partial \check{y}} - \frac{\partial g(y)}{\partial y})$. Because $A$ is a binary attribute, we have

$$a\check{a} = a_1 a_2 > 0.$$

From the property of $g$ that $g(\check{y})$ is strictly convex, we have

$$(y - \check{y}) \left( \frac{\partial g(\check{y})}{\partial \check{y}} - \frac{\partial g(y)}{\partial y} \right) < 0.$$

Note that the derivative of $g(\check{y})$ is $K$-Lipschitz continuous,

$$\left| \frac{\partial g(\check{y})}{\partial \check{y}} - \frac{\partial g(y)}{\partial y} \right| < \frac{2|\check{y} - y|}{\eta(a\check{a}||w \odot \alpha||_2^2 + \gamma^2)},$$

we have

$$\left| \eta(a\check{a}||w \odot \alpha||^2 + \gamma^2)(\frac{\partial g(\check{y})}{\partial \check{y}} - \frac{\partial g(y)}{\partial y}) \right| < 2|y - \check{y}|.$$

Therefore, for every $u$ sampled from the conditional distribution, $|\check{y}' - y'| < |\check{y} - y|$. So we proved

$$\Pr(\{|Y'_{A \leftarrow a}(U) - Y'_{A \leftarrow \check{a}}(U)| < |Y_{A \leftarrow a}(U) - Y_{A \leftarrow \check{a}}(U)|\}|X = x, A = a) = 1.$$

$\square$

## A.5 Proof of Theorem 6.1

*Proof.* For any given $x, a$ we can find the consitional distribution $U_X | X = x, A = a$ and $U_Y | X = x, A = a$ based on causal model $\mathcal{M}$. Consider sample $u = [u_X, u_Y]$ drawn from this conditional distribution. For this sample, we have,

$$\check{x}_{\mathcal{P}_{\mathcal{G}_A}} = \alpha_{\mathcal{P}_{\mathcal{G}_A}} \odot u_{X_{\mathcal{P}_{\mathcal{G}_A}}} + \beta_{\mathcal{P}_{\mathcal{G}_A}} \check{a},$$

$$\check{y}_{PD} = w_{\mathcal{P}_{\mathcal{G}_A}}^{\mathrm{T}} \check{x}_{\mathcal{P}_{\mathcal{G}_A}} + w_{\mathcal{P}_{\mathcal{G}_A}^c}^{\mathrm{T}} x_{\mathcal{P}_{\mathcal{G}_A}^c} + \gamma u_Y.$$

So, the gradient of $g(\check{y}_{PD}, u_{X_{\mathcal{P}_{\mathcal{G}_A}}}, u_{X_{\mathcal{P}_{\mathcal{G}_A}^c}}, u_Y)$ w.r.t. $u_{X_{\mathcal{P}_{\mathcal{G}_A}}}, u_{X_{\mathcal{P}_{\mathcal{G}_A}^c}}, u_Y$ are

$$\nabla_{u_{X_{\mathcal{P}_{\mathcal{G}_A}}}} g = \frac{\partial g(\check{y}_{PD}, u_{X_{\mathcal{P}_{\mathcal{G}_A}}}, u_{X_{\mathcal{P}_{\mathcal{G}_A}^c}}, u_Y)}{\partial u_{X_{\mathcal{P}_{\mathcal{G}_A}}}} + \frac{\partial g(\check{y}_{PD}, u_{X_{\mathcal{P}_{\mathcal{G}_A}}}, u_{X_{\mathcal{P}_{\mathcal{G}_A}^c}}, u_Y)}{\partial \check{y}} \odot w_{\mathcal{P}_{\mathcal{G}_A}} \odot \alpha_{\mathcal{P}_{\mathcal{G}_A}},$$

$$\nabla_{u_{X_{\mathcal{P}_{\mathcal{G}_A}^c}}} g = \frac{\partial g(\check{y}_{PD}, u_{X_{\mathcal{P}_{\mathcal{G}_A}}}, u_{X_{\mathcal{P}_{\mathcal{G}_A}^c}}, u_Y)}{\partial u_{X_{\mathcal{P}_{\mathcal{G}_A}^c}}} + \frac{\partial g(\check{y}_{PD}, u_{X_{\mathcal{P}_{\mathcal{G}_A}}}, u_{X_{\mathcal{P}_{\mathcal{G}_A}^c}}, u_Y)}{\partial \check{y}} \odot w_{\mathcal{P}_{\mathcal{G}_A}^c} \odot \alpha_{\mathcal{P}_{\mathcal{G}_A}^c},$$

$$\nabla_{u_Y} g = \frac{\partial g(\check{y}, u_{X_{\mathcal{P}_{\mathcal{G}_A}}}, u_{X_{\mathcal{P}_{\mathcal{G}_A}^c}}, u_Y)}{\partial u_Y} + \frac{\partial g(\check{y}, u_{X_{\mathcal{P}_{\mathcal{G}_A}}}, u_{X_{\mathcal{P}_{\mathcal{G}_A}^c}}, u_Y)}{\partial y \check{y}_{PD}} \gamma.$$

The response function $r$ is defined as

$$u'_X = u_X + \nabla_{u_X} g, \quad u'_Y = u_Y + \nabla_{u_Y} g.$$

$y'$ can be calculated using response $r$ as follows,

$$y' = y + \eta w_{\mathcal{P}_{\mathcal{G}_A}}^{\mathrm{T}} \left( \alpha \odot \frac{\partial g(\check{y}_{PD}, u_{X_{\mathcal{P}_{\mathcal{G}_A}}}, u_{X_{\mathcal{P}_{\mathcal{G}_A}^c}}, u_Y)}{\partial u_{X_{\mathcal{P}_{\mathcal{G}_A}}}} \right) + \eta w_{\mathcal{P}_{\mathcal{G}_A}^c}^{\mathrm{T}} \left( \alpha \odot \frac{\partial g(\check{y}_{PD}, u_{X_{\mathcal{P}_{\mathcal{G}_A}}}, u_{X_{\mathcal{P}_{\mathcal{G}_A}^c}}, u_Y)}{\partial u_{X_{\mathcal{P}_{\mathcal{G}_A}^c}}} \right)$$

$$+ \eta \left\| w_{\mathcal{P}_{\mathcal{G}_A}} \odot \alpha_{\mathcal{P}_{\mathcal{G}_A}} \right\|_2^2 \frac{\partial g(\check{y}_{PD}, u_{X_{\mathcal{P}_{\mathcal{G}_A}}}, u_{X_{\mathcal{P}_{\mathcal{G}_A}^c}}, u_Y)}{\partial \check{y}_{PD}} + \eta \left\| w_{\mathcal{P}_{\mathcal{G}_A}^c} \odot \alpha_{\mathcal{P}_{\mathcal{G}_A}^c} \right\|_2^2 \frac{\partial g(\check{y}_{PD}, u_{X_{\mathcal{P}_{\mathcal{G}_A}}}, u_{X_{\mathcal{P}_{\mathcal{G}_A}^c}}, u_Y)}{\partial \check{y}_{PD}}$$

$$+ \eta \gamma \frac{\partial g(\check{y}_{PD}, u_{X_{\mathcal{P}_{\mathcal{G}_A}}}, u_Y, u_Y)}{\partial u_Y} + \eta \gamma^2 \frac{\partial g(\check{y}_{PD}, u_{X_{\mathcal{P}_{\mathcal{G}_A}}}, u_Y, u_Y)}{\partial \check{y}_{PD}}.$$

Similarly, we can calculate path-dependent counterfactual value $\check{y}'_{PD}$ as follows,

$$\check{y}'_{PD} = \check{y}_{PD} + \eta w_{\mathcal{P}_{\mathcal{G}_A}}^{\mathrm{T}} \left( \alpha \odot \frac{\partial g(y, u_{X_{\mathcal{P}_{\mathcal{G}_A}}}, u_{X_{\mathcal{P}_{\mathcal{G}_A}^c}}, u_Y)}{\partial u_{X_{\mathcal{P}_{\mathcal{G}_A}}}} \right) + \eta w_{\mathcal{P}_{\mathcal{G}_A}^c}^{\mathrm{T}} \left( \alpha \odot \frac{\partial g(y, u_{X_{\mathcal{P}_{\mathcal{G}_A}}}, u_{X_{\mathcal{P}_{\mathcal{G}_A}^c}}, u_Y)}{\partial u_{X_{\mathcal{P}_{\mathcal{G}_A}^c}}} \right)$$

$$+ \eta \left\| w_{\mathcal{P}_{\mathcal{G}_A}} \odot \alpha_{\mathcal{P}_{\mathcal{G}_A}} \right\|_2^2 \frac{\partial g(y, u_{X_{\mathcal{P}_{\mathcal{G}_A}}}, u_{X_{\mathcal{P}_{\mathcal{G}_A}^c}}, u_Y)}{\partial y} + \eta \left\| w_{\mathcal{P}_{\mathcal{G}_A}^c} \odot \alpha_{\mathcal{P}_{\mathcal{G}_A}^c} \right\|_2^2 \frac{\partial g(y, u_{X_{\mathcal{P}_{\mathcal{G}_A}}}, u_{X_{\mathcal{P}_{\mathcal{G}_A}^c}}, u_Y)}{\partial y}$$

$$+ \eta \gamma \frac{\partial g(y, u_{X_{\mathcal{P}_{\mathcal{G}_A}}}, u_Y, u_Y)}{\partial u_Y} + \eta \gamma^2 \frac{\partial g(y, u_{X_{\mathcal{P}_{\mathcal{G}_A}}}, u_Y, u_Y)}{\partial y}.$$

Thus,

$$|\check{y}'_{PD} - y'| =$$

$$\left| \check{y}_{PD} - y + \eta \left( \|w_{\mathcal{P}_{\mathcal{G}_A}} \odot \alpha_{\mathcal{P}_{\mathcal{G}_A}}\|_2^2 + \|w_{\mathcal{P}_{\mathcal{G}_A}^c} \odot \alpha_{\mathcal{P}_{\mathcal{G}_A}^c}\|_2^2 + \gamma^2 \right) \left( \frac{\partial g(y, u_{X_{\mathcal{P}_{\mathcal{G}_A}}}, u_{X_{\mathcal{P}_{\mathcal{G}_A}^c}}, u_Y)}{\partial y} - \frac{\partial g(\check{y}_{PD}, u_{X_{\mathcal{P}_{\mathcal{G}_A}}}, u_{X_{\mathcal{P}_{\mathcal{G}_A}^c}}, u_Y)}{\partial \check{y}_{PD}} \right) \right|.$$

Denote $p_1 = \frac{1}{2\eta(\|w_{\mathcal{P}_{\mathcal{G}_A}} \odot \alpha_{\mathcal{P}_{\mathcal{G}_A}}\|_2^2 + \|w_{\mathcal{P}_{\mathcal{G}_A}^c} \odot \alpha_{\mathcal{P}_{\mathcal{G}_A}^c}\|_2^2 + \gamma^2)}$. Since the partial gradient of $g(\check{y}_{PD}, u_{X_{\mathcal{P}_{\mathcal{G}_A}}}, u_{X_{\mathcal{P}_{\mathcal{G}_A}^c}}, u_Y)$ w.r.t. $\check{y}_{PD}$ is $2p_1 \check{y}_{PD} + p_2$, we know that $|y' - \check{y}'_{PD}| = 0$. Since for any realization of $u$, the equation holds, we can conclude that the path-dependent LCF holds. □

### A.6 Proof of Theorem 4.1

*Proof.* Since $Y$ is determined by $U$, we denote the causal function from $U$ to $A$ as

$$Y = f(U, A).$$

Suppose the conditional distribution of $U$ given $X = x, A = a$ could denoted as $P_c(U)$, we have

$$\Pr(Y_{A \leftarrow a}(U)|X = x, A = a) = \sum_{u \in \{u|f(u,a)\} = y} P_c(u).$$

Because

$$\Pr(Y_{A \leftarrow a}(U) = y|X = x, A = a) \neq \Pr(Y_{A \leftarrow \check{a}}(U) = y|X = x, A = a),$$

we have,

$$\sum_{u \in \{u|f(u,a) = y\}} P_c(u) \quad \neq \quad \sum_{u \in \{u|f(u,\check{a}) = y\}} P_c(u). \tag{20}$$

Because the predictor satisfies CF,

$$U'_{A \leftarrow a} = r(U, \hat{Y}),$$

$$U'_{A \leftarrow \check{a}} = r(U, \hat{Y}),$$

the future outcome could be written as

$$\Pr(Y'_{A \leftarrow a}(U)|X = x, A = a) = \sum_{u \in \{u|f(r(u,\hat{y}),a) = y\}} P_c(u).$$

From Eq.20, we have

$$\sum_{u|\{f(r(u,\hat{y}),a) = y\}} P_c(u) \neq \sum_{u|\{f(r(u,\hat{y}),\check{a}) = y\}} P_c(u),$$

which is to say

$$\Pr(Y'_{A \leftarrow a}(U)|X = x, A = a) \neq \Pr(Y'_{A \leftarrow \check{a}}(U) = y|X = x, A = a).$$

$$\square$$

## B Parameters for Synthetic Data Simulation

When generating the synthetic data, we used

$$\alpha = \begin{bmatrix} 0.37454012 \\ 0.95071431 \\ 0.73199394 \\ 0.59865848 \\ 0.15601864 \\ 0.15599452 \\ 0.05808361 \\ 0.86617615 \\ 0.60111501 \\ 0.70807258 \end{bmatrix} ; \quad \beta = \begin{bmatrix} 0.02058449 \\ 0.96990985 \\ 0.83244264 \\ 0.21233911 \\ 0.18182497 \\ 0.18340451 \\ 0.30424224 \\ 0.52475643 \\ 0.43194502 \\ 0.29122914 \end{bmatrix} ; \quad w = \begin{bmatrix} 0.61185289 \\ 0.13949386 \\ 0.29214465 \\ 0.36636184 \\ 0.45606998 \\ 0.78517596 \\ 0.19967378 \\ 0.51423444 \\ 0.59241457 \\ 0.04645041 \end{bmatrix} ; \quad \gamma = 0.60754485$$

These values are generated randomly.

## C   Empirical Evaluation of Theorem 5.2

In this section, we use the same synthetic dataset generated in Section 7.1 to validate Theorem 5.2. We keep all the experimental settings as the same as Section 7.1, but use a different predictor $g(\check{y}, u)$. The form of $g(\check{y}, u)$ is

$$g(\check{y}, u) = p_1 \check{y}^{1.5} + p_2 \check{y} + h(u),$$

with $p_1 = \frac{T}{2}$. It is obvious that $g$ satisfies property (i) and (ii) in Theorem 5.2. Because in the synthetic causal model, $y$ is always larger than 0, property (iii) is also satisfied.

Table 3: Results on Synthetic Data for Theorem 5.2: comparison with two baselines, unfair predictor (UF) and counterfactual fair predictor (CF), in terms of accuracy (MSE) and lookahead counterfactual fairness (AFCE, UIR).

| Method | MSE | AFCE | UIR |
|--------|-----|------|-----|
| UF | $0.036 \pm 0.003$ | $1.296 \pm 0.000$ | $0\% \pm 0$ |
| CF | $0.520 \pm 0.045$ | $1.296 \pm 0.000$ | $0\% \pm 0$ |
| Ours | $0.064 \pm 0.001$ | $0.930 \pm 0.001$ | $28.2\% \pm 0$ |

Table 3 displays the results of our method compared to the baselines. Except for our method, UF and CF baselines cannot improve LCF. When the properties in Theorem 5.2 are all satisfied, our method can guarantee an improvement of LCF.

## D   Empirical Evaluation of Theorem 5.3

We generate a synthetic dataset with the structural function:

$$Y = (\alpha U + e^A)^{\frac{2}{3}}.$$

The domain of $U$ is $(0, 1)$. $\alpha$ is sampled from a uniform distribution and set as $0.5987$ in this experiment. In this case, $\tilde{f}(s) = s^{\frac{2}{3}}$. Therefore, property (i), (ii) are satisfied. Since $s > e$, we have $M = \frac{1}{9} e^{-\frac{2}{3}}$. We use a predictor

$$g(\check{y}) = p_1 \check{y}^2 + p_2 + h(u),$$

and choose $u = \frac{1}{2\eta M}$. Table 4 displays the results. Our method achieves a large improvement in LCF.

Table 4: Results on Synthetic Data for Theorem 5.3: comparison with two baselines, unfair predictor (UF) and counterfactual fair predictor (CF), in terms of accuracy (MSE) and lookahead counterfactual fairness (AFCE, UIR).

| Method | MSE | AFCE | UIR |
|--------|-----|------|-----|
| UF | $0.012 \pm 0.001$ | $1.084 \pm 0.000$ | $0\% \pm 0$ |
| CF | $0.329 \pm 0.019$ | $0.932 \pm 0.007$ | $14.0\% \pm 0.65\%$ |
| Ours | $5.298 \pm 1.704$ | $0.124 \pm 0.086$ | $88.6\% \pm 7.93\%$ |

It should be noticed that although CF predictor improves AFCE, it is not contradictory to Theorem 4.1 because Theorem 4.1 is for LCF not relaxted LCF.

# E   Empirical Evaluation of Theorem 5.4

We generate a synthetic dataset following the structural functions 10. We choose $a_1 = 1$, $a_2 = 2$ and $d = 10$. We generated 1000 data samples. The parameters used in the structural functions are displayed as follows.

$$
\alpha = \begin{bmatrix} 0.37454012 \\ 0.95071431 \\ 0.73199394 \\ 0.59865848 \\ 0.15601864 \\ 0.15599452 \\ 0.05808361 \\ 0.86617615 \\ 0.60111501 \\ 0.70807258 \end{bmatrix} ; \quad
\beta = \begin{bmatrix} 0.02058449 \\ 0.96990985 \\ 0.83244264 \\ 0.21233911 \\ 0.18182497 \\ 0.18340451 \\ 0.30424224 \\ 0.52475643 \\ 0.43194502 \\ 0.29122914 \end{bmatrix} ; \quad
w = \begin{bmatrix} 0.61185289 \\ 0.13949386 \\ 0.29214465 \\ 0.36636184 \\ 0.45606998 \\ 0.78517596 \\ 0.19967378 \\ 0.51423444 \\ 0.59241457 \\ 0.04645041 \end{bmatrix} ; \quad
\gamma = 0.60754485
$$

To construct a predictor $g(\check{y})$ satisfies property (ii) and (iii) described in Theorem 5.4, we set

Table 5: Results on Synthetic Data for Theorem 5.4: comparison with two baselines, unfair predictor (UF) and counterfactual fair predictor (CF), in terms of accuracy (MSE) and lookahead counterfactual fairness (AFCE, UIR).

| Method | MSE | AFCE | UIR |
|--------|-----|------|-----|
| UF | $0.036 \pm 0.002$ | $17.480 \pm 0.494$ | $0\% \pm 0$ |
| CF | $1.400 \pm 0.098$ | $11.193 \pm 1.019$ | $35.9\% \pm 11.6\%$ |
| Ours | $1.068 \pm 0.432$ | $0.000 \pm 0.000$ | $100\% \pm 0$ |

$$g(\check{y}) = p_1 \check{y}^2 + p_2 \check{y} + p_3,$$

with $p_1 = \frac{1}{2\eta(a_1 a_2 \|w \odot \alpha\|_2^2 + \gamma_2)}$. Table 5 displays the experiment results. In this case, CF improved LCF. However, we know that there is no guarantee that CF predictor will always improve LCF. And our method, not only can provide the theoretical guarantee, but also achieves a better MSE-LCF trade-off.

# F   Empirical Evaluation On Real-world (Loan) Dataset

We measure the performance of our proposed method using Loan Prediction Problem Dataset (kag). In this experiment, the objective is to forecast the Loan Amount ($Y$) of individuals using their Gender ($A$), income ($X_1$), co-applicant income ($X_2$), married status ($X_3$) and area of the owned property ($X_4$).

The causal model behind the dataset is that there exists an exogenous variable $U$ that represents the hidden financial status of the person. The structural functions are given as

$$
\begin{aligned}
X_1 &= \mathcal{N}(w_1^A A + w_1^U U + w_1^3 X_3 + + w_1^4 X_4 b_1, \sigma_1), \\
X_2 &= \mathcal{N}(w_2^A A + w_2^U U + w_2^3 X_3 + + w_2^4 X_4 b_2, \sigma_2), \\
Y &= \mathcal{N}(w_Y^A A + w_Y^U U + w_Y^3 X_3 + + w_Y^4 X_4 b_Y, 1).
\end{aligned}
$$

$X_3$ and $X_4$ have no parent nodes. We use the same implementation as what we use in the experiments for the Law School Success dataset. Table 6 shows the results of our method compared to the baselines. Again, our method can achieve perfect LCF by setting $p_1$ as $\frac{T}{2}$. Compared to CF predictor, our method has only a slightly larger MSE, but our LCF is greatly improved.

Table 6: Results on Loan Prediction Datset: comparison with two baselines, unfair predictor (UF) and counterfactual fair predictor (CF), in terms of accuracy (MSE) and lookahead counterfactual fairness (AFCE, UIR).

| Method | MSE ($\times 10^4$) | AFCE | UIR |
|---|---|---|---|
| UF | $1.352 \pm 0.835$ | $11.751 \pm 0.848$ | $3.49\% \pm 7.19\%$ |
| CF | $2.596 \pm 0.255$ | $11.792 \pm 0.790$ | $0\% \pm 0$ |
| Ours ($p_1 = T/4$) | $2.646 \pm 0.540$ | $5.896 \pm 0.395$ | $50\% \pm 3.35\%$ |
| Ours ($p_1 = T/2$) | $2.733 \pm 0.197$ | $0.001 \pm 0.000$ | $100\% \pm 0$ |

