# OpenReview forum: "Lookahead Counterfactual Fairness"
_TMLR — Accepted by TMLR_

### Review · Reviewer_KqmR · 2024-09-27

**Summary Of Contributions:**

The paper formulates Lookahead Counterfactual Fairness (LCF) as the objective of achieving counterfactual fairness relative to a downstream outcome of a prediction. They show that achieving counterfactual fairness of a prediction (CF) does not guarantee LCF. Methods for achieving CF typically rely on making predictions based on unobserved features that are not caused by any of the observed features. Even if such features that are predictive for the prediction task exist, they might be causally linked in the causal model of the downstream outcome. Therefore, they cannot be directly used to build LCF predictors.

The paper proposes new methods for achieving LCF by modeling the causal relationships in the form of structural equations and finds predictors satisfying LCF for linear causal models. The causal relationships model a setting where the prediction influences the downstream outcome by causing individuals to strategically respond to the prediction to maximize their chances of receiving a favorable prediction.

The paper implements and evaluates an algorithm that entails estimating parameters of the proposed predictor in synthetic and a real dataset.

**Audience:**

Yes

**Claims And Evidence:**

Yes

**Requested Changes:**

[Critical] Clarification on the statement of Thm 3.1 and its proof: Is the statement that under the conditions of the theorem, any predictor that satisfies CF violates LCF. Or is it that there is a predictor satisfying CF but violating LCF?

**Strengths And Weaknesses:**

**Strengths**
I think the paper identifies a natural form of counterfactual fairness requirement that cannot be met through previous formulations and that require new methods to be satisfied.

The proposed method seems to achieve a measurably higher LCF in the real dataset they consider, compared to previous baselines.

The model of how the prediction affects downstream outcome seems natural and captures the widely studied strategic classification setting.

**Weaknesses**
The proof for Thm 3.1 is missing. And the statement is unclear. It currently seems to imply that under the conditions stated, any predictor that satisfies CF violates LCF. But it seems like it should be that there is a predictor satisfying CF but violating LCF.

The notion of relaxed LCF (Def 4.2) does not seem to be well justified. It seems like it could favor predictors that violate CF by a lot without improving on LCF. One of the methods satisfying LCF (given by Cor 4.1) is empirically evaluated, but justification or evaluation for the other methods (Thm 4.2, 4.3, 4.4) are not provided.

---

> ### Author Response · Authors · 2024-10-21
>
> * Weakness 1: The statement of Theorem 3.1(now Theorem 4.1) is that under the conditions stated, any predictor that satisfies CF violates LCF. We added the proof in the appendix A.6 for the revised edition.
>
> * Weakness 2: The definition 4.2 (now definition 5.2) of Relaxed LCF is not related to CF. Because CF is a definition over the fairness of current predictions. $|Y_{A\leftarrow a}(U) - Y_{A\leftarrow \check{a}}(U)|$ is not CF but the difference between current true label and its associated counterfactual label. It cannot be affected by ML models because it has been determined by the causal model.
>
>      To justify and evaluate scenarios of Theorem 4.2, 4.3, and 4.4 (Theorem 5.2, 5.3, 5.4), we added experiments in Appendix C, D, E of the revised edition.
>
> * Changes 1: Theorem 3.1 (now Theorem 4.1) implies under certain conditions, any predictor satisfying CF violates LCF. We included a polished version of the theorem and its proof in the revised edition.

---

### Review · Reviewer_zW8N · 2024-09-27

**Summary Of Contributions:**

This work proposes the notion of lookahead counterfactual fairness (LCF), a new fairness notion that builds upon counterfactual fairness with a specific constraint of requiring fairness in future outcomes. The idea is motivated by the situation where predicted outcomes may have downstream effects for future outcomes (e.g., bank loans may decrease credit scores that can determine future loan approvals). The authors define conditions under which a predictor satisfies LCF and perform experiments on synthetic and real-world datasets to demonstrate the effectiveness of the proposed notion in ensuring counterfactually fair future outcomes.

**Audience:**

Yes

**Claims And Evidence:**

No

**Requested Changes:**

## Requested Changes

Critical

- It would be good to see experiments on more real-world datasets where the CF predictor produces a large disparity in the future outcome, but the LCF predictor has a clear advantage.
- How is the value of $p_1$ derived? The authors define it w.r.t the parameters of the structural equations, but there is no intuition or derivation of this value. This is important since this value has a large influence on achieving the proposed notion of LCF.
- I strongly suggest the authors revise all the proofs (and include the proof for Theorem 3.1). There are several errors in the provided proofs in the appendix

    In A.1:

    1. I am not sure that the gradients in Eq 13 and 14 are computed correctly. How exactly is this computed?
    2. The response function $r$ is not defined and made clear for each proof.
    3. In Eq. 21, the derivative of g w.r.t $\widehat{y}$ is not computed correctly. Instead of $2p_1\widehat{y}$, it should be $2p_1\widehat{y} + p_2$.
    4. Corollary 4.2 should be Corollary 4.1
    5. How do Eq. 19 and 21 imply Eq. 22? There seems to be a big leap in this step.

    Generally:
    1. It would be helpful to fully define the predictor g in each of the proofs before computing the gradients.


Nice to have

- The notion of lookahead counterfactual fairness is similar to the idea of long-term fairness from Hu and Zhang (2021). It would be good to elaborate on the relationship between the two approaches. It seems to me that Hu and Zhang focus on *long-term fairness* in the sequential decision-making regime. However, the formulation is not with respect to counterfactual fairness.
- I think a general proof sketch of each theorem would be beneficial to get an idea of why the assumptions are important to prove the result.

**Strengths And Weaknesses:**

## Strengths

- Overall, I think the general notion of lookahead counterfactual fairness is interesting and has strong real-world motivation.
- The paper is well written with good examples to illustrate the intuitions behind the proposed notion.
- The extension to path-specific lookahead counterfactual fairness increases the utility of the approach for discrimination following indirect causal paths in future outcomes.

## Weaknesses

- The paper lacks contextualization with other notions of causal fairness. I believe it would be beneficial to include a related works section and compare the proposed definition with other fairness notions and their goals such as Kusner et al (2017), Hu and Zhang (2021), etc.
- It is not clear to me why Theorem 3.1 holds. What is the intuition behind why a counterfactual fairness constraint violates the LCF definition? There is no proof of this theorem provided in the appendix.
- In the empirical evaluation, the two datasets are quite different in the produced result. In the synthetic data case, it is evident that the CF predictor induces a large disparity between the factual and counterfactual future outcomes. However, in the law school dataset experiments, the CF already seems to perform quite well since the distributions are very close. In this scenario, why is LCF a better constraint than CF for future outcomes? I am not completely convinced that the proposed algorithm is desirable as opposed to CF from the current experimental results.
- The proofs are hard to read as there are many errors and inconsistencies. I don’t believe the gradients are computed correctly at several places in the proof.

---

> ### Author Response · Authors · 2024-10-21
>
> * Weakness 1: Thank you for your comment. We have the discussion about the notion “counterfactual fairness” proposed by Kusner et al. (2017) in Section 2.2 (Now section 3.2), and Hu and Zhang (2021) in introduction of the first submission. We added a separate related work section (Section 2) in the revised edition to discuss several causal fairness notions.
>
> * Weakness 2: We included the proof for Theorem 3.1 (Now theorem 4.1) in Appendix A. 6 in the revised edition. The intuition behind the theorem is that if the condition in the theorem holds, then a counterfactually fair predictor results in the same $U’$ in the factual and counterfactual world. Given a simple example of causal structural functions like Y’ = U’ + A, we know that the difference of the true labels between the factual and counterfactual world will violate LCF.
>
> * Weakness 3: When the distribution of $Y$ in the factual world and counterfactual world is significantly different, this difference will not be mitigated after the response to a CF predictor just like the example we explained for weakness 2.  The reason that the difference is not obvious in the Law school dataset is because the distribution of $Y$ in the factual world and counterfactual world are not significantly different. However, this is not the case in our synthetic data experiment.
>
> * Weakness 4: We checked all the proof in the paper and modified the typos and inconsistencies. They do not affect the flow of the proof and the conclusion. And to make the proof easier to read, we provide proof sketches for the theorems in the revised edition.
>
> * Changes 1: We added a real-world experiment in the revised edition in Appendix F.
>
> * Changes 2: We provide an example to explain the value of $p_{1}$ here. Consider an example where business owners apply for loans to develop their companies. Let features $X$ be the net value of each company, sensitive attribute $A$ be the gender of business owner, and label $Y$ be the owner’s actual ability to repay the loan. The bank, by training a model to predict $Y$, decides the amount of loan $\hat{Y}$ to issue to the owner. Suppose there is a latent variable $U_{X}$ representing the unknown resources that affect the owner’s repayment ability but irrespective of gender. How much loan the bank issued to the owner $\hat{Y}$ will change the resources the owner has and affect the net value in the future $X'$ as well as the future repayment ability $Y’$.
> For the above example, consider a case where the owner in factual world is man and a woman in counterfactual world. Suppose net value $X = \alpha U_{Y} + \beta A$ is lower for women in the counterfactual world, and repayment ability $Y = wX + \gamma U_{Y}$. To ensure the owner in factual and counterfactual worlds to have the identify repayment ability in the future, the bank needs to issue more loans to women to bridge the gap. $p_{1} = \frac{2}{\eta (||w \odot \alpha||_{2}^{2} + \gamma^{2})}$ is used to control how much more money we give the woman such that gap will be just bridged while not giving too much money to make the woman privileged than the man.
>
> * Changes 3:
>  * 1. We explained how the gradients in Eq 13 and Eq 14 are computed in detail in Appendix A.1 of the revised edition. Because $g(\check{y}, u_{X}, u_{Y})$ is a function of $u_{X}$, $u_{Y}$ and $\check{y}$ and $\check{y}$ itself is a function of $u_{X}$ and $u_{Y}$, the gradients of $g$ w.r.t. $u_{X}$ and $u_{Y}$ can be computed with the chain rule.
>  * 2. We added an explanation of $r$ to every proof in Appendix A to make it clearer.
>  * 3. Thank you for pointing out the mistake. We revised the error as well as others in the new edition. Because $p_{2}$ will be cancelled out when computing $|y' - \check{y}'|$, it will not affect our conclusion.
>  * 4. Thank you for pointing out the inconsistencies. We revised them in Appendix A.1 of the new edition.
>  * 5. To get Eq 22 from Eq 19 and Eq 21, we replace $p_{1}$ by $\frac{T}{2}$ in Eq 21 and then we replace derivative of $g$ in Eq 19 by Eq 21.  This results in Eq 22. We explained the detailed process in Appendix A. 1 of the revised edition.
>  * 6. Thank you for your suggestions. We made the modifications in Appendix A.1 to make the proof clearer.

---

> ### Author Response · Authors · 2024-10-21
>
> * Changes 4: The main difference between Hu and Zhang (2021) and our work is that they are trying to ensure the fairness of future predictions. We want to make the future true labels become fair. We clarified this in the introduction and added more explanations to the related work section in the revised edition.
>
> * Changes 5: Thank you for your suggestion. We added the proof sketch in the main paper following every theorem in the new edition.

---

### Review · Reviewer_wQHn · 2024-10-08

**Summary Of Contributions:**

This paper introduces lookahead counterfactual fairness (LCF), which requires parity over future outcomes between the factual and counterfactual worlds for a decision to be considered “fair”. The paper characterizes when such parity is possible under simple structural causal models, and how to find a LCF-fair predictor.

**Audience:**

Yes

**Broader Impact Concerns:**

Since we never have access to a reasonable structural causal model, counterfactual fairness approaches tend not to be applied to real decisions, limiting concerns about broader impact. To the extent that this paper shifts the discussion towards the consequences of decisions, its broader impact could be positive.

**Claims And Evidence:**

Yes

**Requested Changes:**

## Critical

*Strategic response vs decision consequences*

There are two downstream effects that could be relevant for the fairness of a decision: the consequences of the decisions (eg default and further financial harm as the consequence of issuing a loan that can’t be repaid), and the strategic response to the existence of the decision making system (eg changing behavior to increase the chance of being issued a loan). The former is pretty interesting; it seems reasonable to make decisions that produce similar consequences for the factual and counterfactual decision subjects. However, this paper mostly focuses on the latter effect from Sec 4 onwards. This has some value as an extension of the strategic classification literature, but to me is much less interesting than a notion of fairness that focuses on the first-order consequences of the decisions on people.

Why was the decision made to focus on strategic response? Can the machinery developed in this paper be repurposed to ensure LCF with respect to decisions' consequences?

Relatedly, why is the future outcome the same as the predicted outcome? This makes sense if we only care about strategic responses, but the paper is motivated by the need to consider the consequences of the decisions. In the lending context, surely the future outcome of interest is some sort of financial wellbeing measure? Whether the applicant would be creditworthy for a subsequent loan is of course related to their future financial state, but it’s pretty tangentially related compared to something like income or wealth.

If the paper is going to be motivated by the value of considering the consequences of decisions on people, then I think it needs to be changed to consider those consequences more directly. As it stands the paper is really about strategic classification, which is fine (though much less interesting to me) but not made clear in the early sections.

## Smaller edits/clarifications

> “But in a counterfactual world where the applicant belongs to another group, he/she is not qualified.”

This needs more explanation. It’s contested as to whether it even makes sense to reason about counterfactuals for immutable characteristics. But even granting that such a thing makes sense, why is this person no longer qualified to receive the loan? Is the mechanism “counterfactually change person to disadvantaged group” -> “person faces education/employment/housing discrimination” -> “person has worse income/wealth” -> “person no longer qualifies”? But if this is what we mean by a counterfactual, why would it be desirable to equalize decisions between counterfactuals as required by CF? For example, it seems obviously undesirable to refuse government support to someone because in the counterfactual world where they hadn’t faced discrimination they would have a higher income, making them ineligible.

**Strengths And Weaknesses:**

## Strengths

Lookahead counterfactual fairness seems strictly better than counterfactual fairness, since we should care more about the outcomes of decisions than the decisions themselves. And since counterfactual fairness is popular, this paper may be of interest to a lot of people in the field.

The paper seems technically sound, though I haven't checked the details.

## Weaknesses

This paper inherits many of the substantial weaknesses of counterfactual fairness:
* the completely implausible assumption that a structural causal model for the system being studied is available
* the dubious notion of counterfactuals for immutable characteristics like race and gender

These weaknesses don't rule this paper out, since the original counterfactual fairness paper shares them and is highly influential. If we accept the counterfactual fairness framework as sound, there are still questions about this paper (see below) that I would want answered before I can recommend it for acceptance.

---

> ### Author Response · Authors · 2024-10-21
>
> * Weakness 1: Thank you for your comment. While we made several assumptions in our work, these assumptions are standard in literature to facilitate counterfactual inference (Kusner et al., Wu et al., Chiappa et al.).
>
> * Changes 1: We want to clarify that in our model, the agents are responding to the decision consequence (i.e., after the decision is being made). However, they respond in a strategic way to improve the outcome in the next round of the decision. This is the reason that the gradient of the decision is impacting future features and qualifications.
>
> * Changes 2: Thank you for bringing up the example about government support. We agree that counterfactual fairness is not a perfect fairness notion and cannot be applied to all scenarios. In particular, the fairness notion should be aligned with scenario that we are considering. On the other hand, the counterfactual fairness is suitable for the college admission scenario provided by Kusner et al. (2017) where the decision should be based on the knowledge and talent (which is unobservable) rather than GPA which is impacted by discrimination.
>
> Kusner, Matt J., et al. "Counterfactual fairness." Advances in neural information processing systems 30 (2017).
>
> Wu, Yongkai, Lu Zhang, and Xintao Wu. "Counterfactual fairness: Unidentification, bound and algorithm." Proceedings of the twenty-eighth international joint conference on Artificial Intelligence. 2019.
>
> Chiappa, Silvia. "Path-specific counterfactual fairness." Proceedings of the AAAI conference on artificial intelligence. Vol. 33. No. 01. 2019.

---

### Review · Reviewer_47kL · 2024-10-08

**Summary Of Contributions:**

The authors address a critical gap in the fairness literature by proposing a fairness notion, Lookahead Counterfactual Fairness (LCF), which accounts for downstream effects of machine learning models on individuals' future outcomes. This work broadens the scope of fairness by integrating long-term consequences into the fairness paradigm. Furthermore, the authors propose a learning method to achieve LCF.

**Audience:**

Yes

**Claims And Evidence:**

No

**Requested Changes:**

1. It would be helpful to include an overview or intuitive explanation for the proof of each theorem, making the theoretical results more accessible.

2. Incorporating more diverse datasets from domains such as healthcare or financial lending could better illustrate the advantages of the proposed methods, particularly in cases where CF underperforms compared to LCF.

3. The predictor g in Eq. 8 is not clear to me. Specifically, why does the input of the predictor include $\check{Y}$ and $U$, given that $U$ is not an observed variable? How can it be used as input for the predictor?

**Strengths And Weaknesses:**

S1. The paper introduces LCF, which considers fairness in long-term effects. This addresses a known limitation in existing fairness frameworks, where ML predictions satisfying fairness in a static sense can still lead to unintended future disparities.

S2. The authors provide a solid theoretical foundation for LCF, identifying conditions under which ML models can satisfy this fairness notion.

S3. The extension to path-dependent LCF is another valuable contribution, adding flexibility to the application of LCF by focusing on specific causal paths that may influence unfair outcomes.

W1. While the paper focuses on linear causal models and provides comprehensive results for these cases, many real-world systems involve complex, non-linear interactions. This is an important area that should have received more attention.

W2. The theoretical guarantees of LCF rely on assumptions (e.g., invertibility of certain functions) that may not always hold in practical settings.

W3. The experiments primarily focus on a synthetic dataset and the Law School dataset. Incorporating more diverse datasets would better demonstrate the proposed fairness framework, particularly since the density plot comparisons for the Law School dataset show minimal differences across methods.

---

> ### Author Response · Authors · 2024-10-21
>
> * Weakness 1: Thank you for your comment about the non-linear case. The LCF definition we proposed is not restricted to the linear causal model. Our discussion about the linear causal model is a starting point to developing LCF-guaranteed algorithms. Theorem 4.3 and Theorem 4.4 (Now Theorem 5.3 and Theorem 5.4 in the revised edition) provides theoretical analyses on how a predictor can ensure LCF when the causal model is non-linear.  We revised Theorem 4.3 (Now theorem 5.3) to satisfy LCF in a more general non-linear case.
>
> * Weakness 2: Thank you for your comment. While we made several assumptions in our work, these assumptions are standard in literature to facilitate counterfactual inference (Kusner et al., Wu et al., Chiappa et al.). For example, the assumption of bijective causal model is widely used to ensure the counterfactual identifiability (Nasr-Esfahany et al.).
>
> * Weakness 3: Thank you for your suggestion. We added an additional experiment on the real-world dataset in Appendix F of the revised edition.
>
> * Changes 1: We added a proof sketch for every theorem in the main paper just following the theorems in the revised edition.
>
> * Changes 2: We included a new experiment using a new real-world dataset in Appendix F of the revised edition.
>
> * Changes 3: $U$ is an unobservable variable but the conditional distribution of $U$ given X = x, A = a can be inferred. Then we follow the same practice as Kusner et al. and we draw sample $u$ from conditional distribution $U|X=x, A=a$ and generate corresponding $\check{y}$ to calculate final prediction.
>
> Nasr-Esfahany, Arash, Mohammad Alizadeh, and Devavrat Shah. "Counterfactual identifiability of bijective causal models." International Conference on Machine Learning. PMLR, 2023.
>
> Wu, Yongkai, Lu Zhang, and Xintao Wu. "Counterfactual fairness: Unidentification, bound and algorithm." Proceedings of the twenty-eighth international joint conference on Artificial Intelligence. 2019.
>
> Chiappa, Silvia. "Path-specific counterfactual fairness." Proceedings of the AAAI conference on artificial intelligence. Vol. 33. No. 01. 2019.
>
> Kusner, Matt J., et al. "Counterfactual fairness." Advances in neural information processing systems 30 (2017).

---

### Decision · Action_Editor_27Gx · 2024-11-19

**Recommendation:** Accept as is

**Comment:**

This paper proposes look ahead counterfactual fairness,  a fairness notion accounting for the downstream effects of machine models which requires the individual to be counterfactually fair. The conditions under which the proposed fairness motion can be satisfied is derived theoretically. An algorithm is also proposed to achieve the proposed fairness notion. The experiments on both synthetic and real data demonstrate the effectiveness of the proposed method. The paper had some flaws but were fixed in the rebuttal phase. The proposed idea is interesting and has the potential to inspire new works on causal fairness. I thus recommend acceptance of this paper.

**Audience:**

Yes.

**Claims And Evidence:**

Yes.